# Effector Complexity Enhances Transfer Learning

## Abstract

Both biological and artificial embodied systems rely on effectors to interact with the world. How does this embodiment impact the way they learn? The role of embodiment in shaping learning dynamics is not well understood from a neuroscience or a machine learning perspective. In this study, we treat embodiment as a variable in artificial agents and study how changes in effector complexity reshape the dynamics of learning. Our hypothesis is that more complex effectors provide constraints that yield better transfer learning on new tasks, despite simultaneously posing a more complex control problem. We evaluated this hypothesis on area under the performance curve, and use time to sustained performance plateau as a parameter for task difficulty. Our results show that while a simpler effector excels when trained from scratch, a more complex effector yields superior performance after pre-training on another task. We further demonstrate that the improvement gained from transfer learning is greater for the complex effector. Our findings suggest that embodiment plays an important role in enabling efficient transfer, offering insights into the differences in learning dynamics between disembodied artificial systems and their biological counterparts.

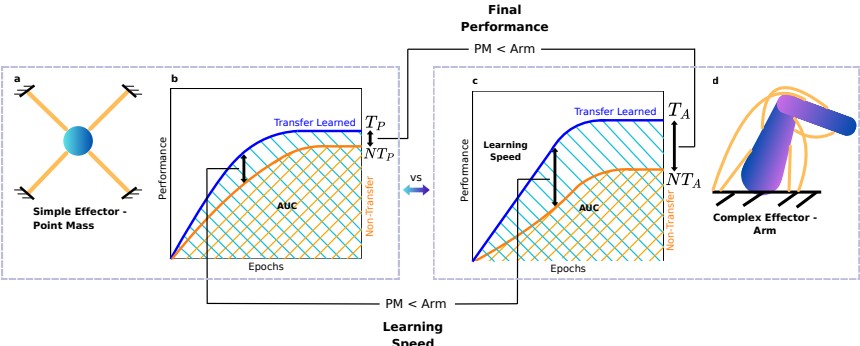

Figure 1: Summary of the Approach and Main Result: Simple effectors learn faster from scratch, but complex effectors achieve greater benefits from transfer.
(a) Simple Effector: A Point Mass (blue) controlled by four diagonal muscles (yellow) moving in the 2D plane.
(b) Point Mass performance: no-transfer (orange) vs transfer (blue). Modest improvement with transfer performance ($T_P > NT_P$) and learning speed, modest AUC.
(c) Arm performance: no-transfer (orange) vs transfer (blue), stronger transfer performance benefits ($T_A >> NT_A$), and learning speed, indicating better knowledge reuse. Overall, larger AUC after transfer.
(d) Complex Effector: A biomechanical Arm with two links (purple) and six muscles (yellow), introducing higher embodiment complexity.

# 1 Introduction

Deep learning based approaches have seen significant success in many domains ranging from image to speech. However, these models usually require large amounts of high-quality data and long training durations to reach competitive performance (Talaei Khoei et al., 2023). Deep Reinforcement Learning (DRL), which combines reinforcement learning with deep neural networks, has shown promise in learning effective function approximations for tackling complex problems (Arulkumaran et al., 2017) but struggle with cross-task adaptation. In contrast, humans can quickly adapt to new tasks by transferring meta-skills acquired from diverse experiences throughout their lives (Lake et al., 2015). This is in sharp contrast to current deep reinforcement learning-based navigation methods, where policy networks are trained from scratch.

Embodiment is the principle that intelligence is not solely computational or abstract, but tightly integrated with the agent's body and physical capabilities (Zhao et al., 2024). It allows a system to have meaningful interactions with the environment it is placed in (Roy et al., 2021). The integration of a physical body and sensory-motor feedback that allows the ability to transfer and reuse knowledge across tasks, should help by reducing the need for extensive training samples (Li et al., 2020). In robotics, perception-based feedback has traditionally dominated, with agents relying on visual inputs to estimate their state and take necessary action (Sünderhauf et al., 2018; Levine et al., 2018; 2016). In this work, we propose a shift toward actuator-based feedback, leveraging the embodiment of effectors, i.e., getting feedback from proprioception instead of depending on perception. We study transfer learning in embodied neural networks, where a model trained on a source task (referred to as the Base Model), is subsequently trained on a target task (referred to as the Transfer Model) using policy reuse. The hypothesis we would like to test is if embodiment improves learning speed and final performance. We assess this hypothesis across a diverse set of tasks and effectors and evaluate transfer quality using four metrics: peak performance, index at peak, area under the performance curve and time to sustained plateau. All biological learning systems are embodied, motivating the hypothesis that embodiment impacts a wide range of functions of the brain including decision making and learning (Cisek & Pastor-Bernier, 2014). However, precisely how embodiment shapes the dynamics of learning in the brain is not well understood. The body is often treated as a fixed substrate instead of one with tunable parameters, and the implications of different body plans remain largely unexplored. By treating embodiment as a variable in artificial agents, we examine how effector complexity impacts the dynamics of learning a new task, both from arbitrary initial conditions and in models with prior training on other tasks.

## 1.1 Related Work

The role of embodiment in learning has received previous attention in several different domains. For example, Kulkarni & Nair (2023) examined the problem of transfer learning in the context of embodied systems with robots and neural network controllers. They propose a new method for transferring knowledge between different tasks using an evolutionary algorithm to train neural networks instead of reinforcement learning. They also demonstrate that 'hot neurons', neurons with significant positive contribution to learning, are particularly important for transfer learning. However, the authors point out the lack of methods to measure such transfers among different tasks. Our study seeks to overcome this problem by introducing metrics to evaluate transfer learning performance systematically.

Embodiment can also impact the performance of language models. For example, Driess et al. (2023) show the capabilities of an embodied language model, PaLM-E by injecting multi-modal information into the embedding space. PaLM-E directly incorporates continuous inputs from sensor modalities into a large language model, enabling more grounded inferences for sequential decision making in the real world. While their focus is on vision guided multi-modal grounding, our work emphasizes actuator feedback as the primary signal providing an alternative approach to embodiment that is not dependent on perception. The authors also demonstrate that when PaLM-E was evaluated on robotic manipulation tasks, multi-task training improves performance compared to training models on individual tasks and also improves data efficiency. As tasks become more complex, the performance of a neural network controller decreases, especially when learning happens through an embodied system (Kulkarni & Nair, 2023).

Recent work suggests that architectural complexity and embodiment can jointly influence how well agents can generalize across tasks. Studies such as those from Reed et al. (2022) and Das et al. (2018) also report performance improvement for multi-modal tasks. In addition to LLM-based models, the work in Lin et al. (2024) also defines the Clip on Wheels method (CoW) (Gadre et al., 2023) which uses zero-shot visual models like CLIP (Contrastive Language–Image Pre-training) (Radford et al., 2021) for embodied AI navigation and relies on a GRU-Actor + GRU-Critic formation for learning. Overall, prior work provides promising evidence that embodiment influences learning dynamics in multi-modal settings, motivating a more thorough investigation into its role in transfer learning.

## 2 METHODS

### 2.1 NETWORK ARCHITECTURE

The gated recurrent unit (GRU) is particularly suited for sequential data applications, such as time-series or trajectory modeling, where adaptive hidden states and robust initialization improve performance. Our architecture includes a custom GRU module that extends PyTorch's *nn.GRU* with a learnable hidden state. The hidden state $hidden_0$ is parameterized and constrained through a $tanh$ activation to remain bounded and numerically stable. This design allows the network to adapt its internal dynamics to the structure of the task from the outset of training. Our work builds on the Motornet library (Codol et al., 2024). For training structure, please see Appendix A.2.

### 2.2 TRAINING

We define baseline learning performance by training each network architecture and effector on every task independently from random initial conditions, without prior experience from other tasks. This 'no-transfer' model serves as a comparison to quantify the benefit of transfer learning.

1. Base model: trained on the source task.
2. Transfer model: initialized with the base model, then trained on a new task.
3. No-Transfer model: counterpart of the transfer model that was trained from scratch.

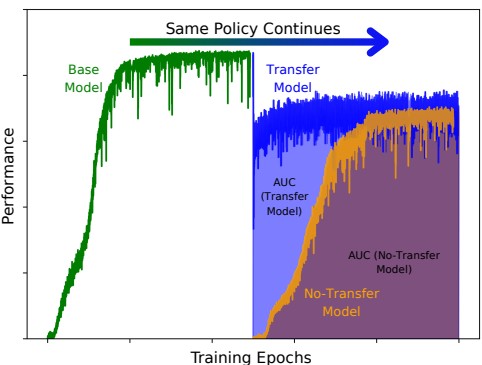

Figure 2: Overall Training: First a Base Model (green) is trained on a task (e.g. RTR). The same model is then trained on a transfer task (e.g. COR), and hence known as a Transfer Model (blue). The Transfer Model is then compared to a No-Transfer Model that is trained on the same transfer-task (i.e. COR) but from scratch.

### 2.3 TASKS

To test our hypothesis and model, we created six different reaching tasks, which are summarized in Table 1. All tasks are restricted to a 2-D plane, where goal locations are in (X,Y) co-ordinates.

Table 1: Task descriptions

| Task Name | Description |
|---|---|
| Random Target Reach (RTR) | Reach randomly generated targets from randomly generated joint states |
| Center Out Reach (COR) | Reach randomly generated targets from Cartesian center joint states |
| Center Out and Back, Fixed Goal Reach (COBFGR) | Reach randomly generated targets from Cartesian center joint states, and return to center. 'Visual' goal remains the same but the 'rewarded' goal changes. This gives the effector mixed signals about returning to the center. |
| Center Out and Back, Jumping Goal Reach (COBJGR) | Reach randomly generated targets from Cartesian center joint states, and return to center. Both 'visual' and 'rewarded' goal shift giving the effector a strong signal to return to the center. |
| Delayed Center Out Reach (DCOR) | Start from the Cartesian center joint states and wait before moving to a randomly generated target. |
| Jumping Target Reach (JTR) | Reach a new randomly generated target every time a target is hit. |

The start configuration of the effector is characterized in joint angles in the case of Arm, or end effector location (X,Y) in the case of Point Mass. Both, goal locations and starting configuration of the effector can be randomized depending on the task.

For all tasks except for JTR, performance was evaluated as the fraction of time the model spends in the target zone on each trial. Thus, because it takes a finite amount of time to move from the start location to the target, optimal performance is a fraction less than 1.0. For example, an average reach takes 15% of a trial in RTR, and COBJGR duplicates this reach, making optimal performance in these tasks approximately 85% and 70%, respectively. For JTR, time spent in the target region is negligible even for expert-level performance, because the target moves upon being reached. Therefore, we quantified performance in JTR as the number of new targets reached on a trial normalized by the number of targets presented during the trial.

We evaluated performance across this family of reaching tasks that have varying initial conditions, goal dynamics, and temporal constraints. These tasks are designed to vary in complexity to probe different aspects of control and adaptability. While some tasks were more difficult than others, these tasks are not organized along a predefined difficulty scale. Instead, we quantify difficulty using Time to Plateau (TTP). By training models on each task from scratch (without transfer) and measuring how long they take to reach stable performance, we obtain an empirical proxy for difficulty. This design allows us to ground our comparisons in data driven metrics rather than qualitative assumptions about complexity (see Table 1). We define TTP as the number of epochs taken for the performance of the No-Transfer models to reach and stabilize near their respective peak values. A Savitzky-Golay filter (Savitzky & Golay (1964), Schafer (2011)) is applied with a window length of 500 and a polynomial order of 3 to smooth out fluctuations in the data. Next an amplitude tolerance is set to 10% of the respective peak value, allowing for slight deviations from the peak to still be considered part of the plateau. A plateau is identified when the slope is below a small threshold ($1e^{-4}$ in this case) and when the performance difference between that point and the peak is within amplitude tolerance. At such an index, a plateau is said to have been established. We show difficulty of each task in Figure 7 (left). The time to sustained plateau ($T_{\text{plateau}}$) is calculated as:

$$T_{\text{plateau}} = \min\left\{ t : |y_t - y_{\text{peak}}| < \epsilon \text{ and } \frac{\Delta y_t}{\Delta t} < \delta \right\}, \quad (1)$$

where:

- $y_t$: Performance value at time $t$.
- $y_{\text{peak}}$: Peak performance value.
- $\epsilon$: Amplitude tolerance threshold for deviation from $y_{\text{peak}}$.
- $\frac{\Delta y_t}{\Delta t}$: Slope of the performance curve.

- $\delta$: Threshold for slope, defining a stabilized plateau.

## 2.4 EFFECTORS

We create the 'embodiment' for the neural network to control through two effectors. These effectors are based on the Motornet library for python Codol et al. (2024) which is a toolbox for building and controlling differentiable biomechanical effectors.

### 2.4.1 POINT MASS

The first effector is a simple point mass object with a central mass connected to four muscles diagonally creating an "X" formation Figure 3(a). One end of each muscle (yellow) is fixed to the ground/ reference frame and the other end is attached to the point mass itself (blue). Each muscle is modeled as a rectified linear muscle who's force output is a linear function of its maximum isometric force and activation value which itself is bound between 0 and 1. The end effector itself is modeled as a bony structure and can be considered as a one-bone system with no joints. However, note that the number of degrees of freedom is not zero like the number of joints, but is two, corresponding to the freedom to move in the X-Y plane.

### 2.4.2 ARM

The Arm is the more complex effector shown in Figure 3(b). It is a two-link, six-muscle model with the 'end effector' being the end point of the second link. It also integrates biomechanical details and approximations for controlling a robotic arm. This effector class is an implementation of a 6-lumped muscle model form Nijhof & Kouwenhoven (2000) since lumped muscles can be considered a functional approximation of biological reality. The skeleton consists of two parts. The first bone (Humerus) between the shoulder and elbow and the second bone (Radius and Ulna, modeled as a single link) between the elbow and the end effector. This system is made to perform as a planar arm with two degrees of freedom. The six muscles represent functional approximations of biological actuators. The skeleton and each muscle has individual physical parameters such as weight, length, tendon length, inertia etc. For further numerical specifications refer to the appendix. The effector uses a third degree polynomial function to approximate the moment arms, musculotendon lengths, and musculotendon velocities during movement. A moment arm describes how changes in joint position affect tendon force transmission while musculotendon refers to the overall muscle + tendons structure. Euler integration is used as the default numerical method and control inputs directly actuate the muscles, influencing joint torques through moment arms.

## 2.5 PERFORMANCE METRICS

Area Under the Curve (AUC) is calculated to quantify the overall performance of the task and baseline models over a defined interval. In this case, AUC is calculated using the trapezoidal rule for numerical integration. It is extracted starting from the end of the baseline and extending till the last epoch $(n)$ of the transfer task. AUC is calculated as:

$$\text{AUC} = \sum_{i=1}^{n} \frac{y_i + y_{i+1}}{2}(x_{i+1} - x_i),\tag{2}$$

where:

- $y_i$ and $y_{i+1}$ are the performance values at indices $i$ and $i + 1$, respectively.
- $x_i$ and $x_{i+1}$ are the corresponding indices or time steps.

## 3 RESULTS

### 3.1 SIMPLER EFFECTORS LEARN FASTER FROM RANDOM INITIAL CONDITIONS

We quantified the dynamics of learning using the area under the curve (AUC) of the performance over training trials. This metric quantifies performance over the entirety of learning without additional parameters. To compare performance of 'No-transfer' models across tasks, we compute

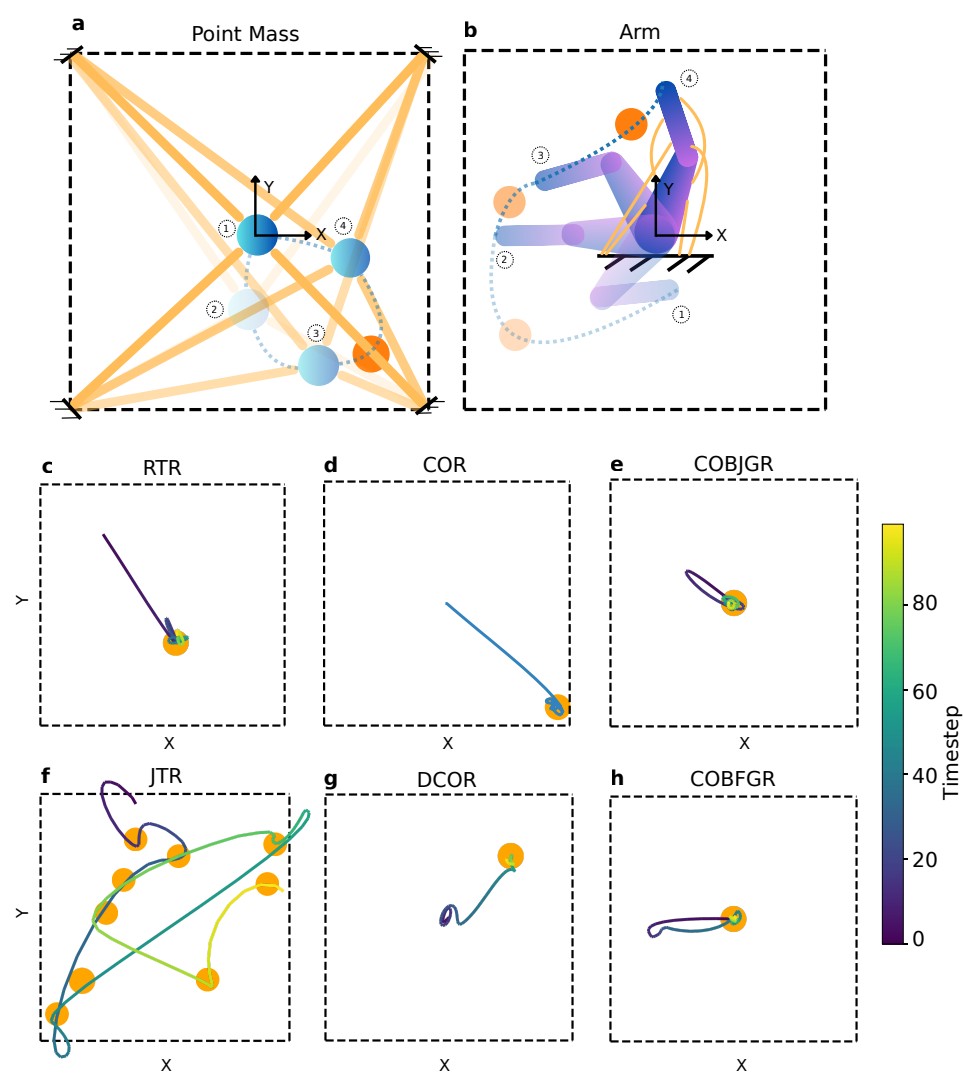

Figure 3: (a) The Point Mass is a central mass (blue), connected by four diagonal muscles. In this case the end effector is the mass itself. An example sequence of the Point Mass performing COBFGR is shown from points 1 through 4, where the dotted line (blue) shows the trajectory progression, the orange circle shows the target. The four muscles move in sequence to pass the Point Mass through the target. Once the target is hit, the Point Mass moves back to the center.

(b) The Arm is built from two links (magenta) and six muscles (yellow). Links move when the muscles contract in sequence. An example sequence of the Arm performing JTR is shown from points 1 through 4, where the dotted line (blue) shows the trajectory progression, the orange circles show the target. The tip of the arm (end effector) must pass through the target to get the reward.

(c) - (f) Different tasks (as described in Table 1) being completed in the X-Y plane, with the end-effector trajectory and target plotted. Each trial lasts 100 time steps.

$AUC_{Arm} - AUC_{PointMass}$. Figure 4 shows examples of this learning trajectory for both effector models. In this No-Transfer Scenario where models start from random initial conditions, the Point Mass model outperformed the Arm model on all tasks as shown in Figure 4 (g). Intuitively, this follows from the fact that simpler effectors are easier to control, leading to faster learning from random

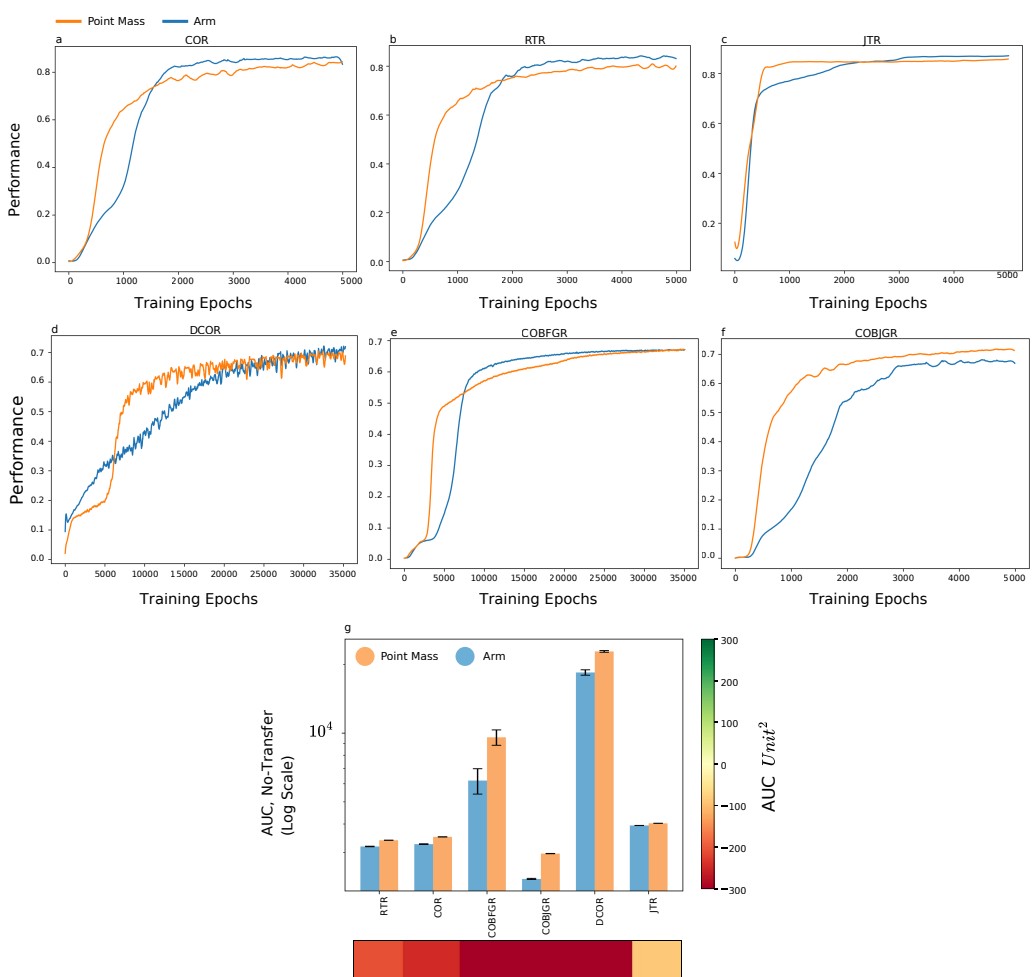

Figure 4: No-transfer models performance comparison for different tasks
(a-f) Training performance of different tasks, in no particular order. Note that the Point Mass (yellow) learns faster, and matches or sometimes, exceeds the Arm's (blue)
final performance.
(g) A bar plot with Log scale Y axis. Shows the relative difference in the AUCs covered, and a heatmap to visualize the difference ($AUC_{Arm:No-transfer} - AUC_{PointMass:No-transfer}$).

initial conditions to expert level performance in each task. This finding directly parallels prior work from Bazzi et al. (2024), which shows that complex manipulation is often substituted by simpler approximations.

## 3.2 INCREASED EFFECTOR COMPLEXITY IMPROVES TRANSFER PERFORMANCE

As described above, we examined transfer learning by taking models previously trained on one task and training them on one of the new tasks (Methods). Intuitively, previous training on one task should confer some benefit to the speed of acquisition of a new task, if the two tasks share some of the same features.

To quantify the improvement in learning due to transfer, we defined a metric $AUC_{Transfer} - AUC_{No-Transfer}$ separately for the arm and point mass. Across all six tasks, transfer models using the Arm showed positive improvements in AUC relative to their respective baselines (Figure 6 (right) and Figure 7 (right)).

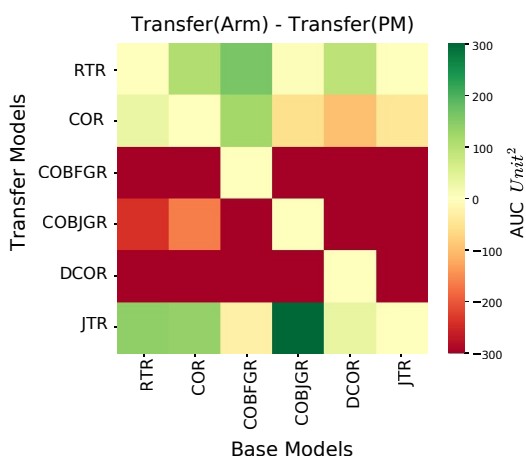

Figure 5: Transfer (Arm) - Transfer (PM)

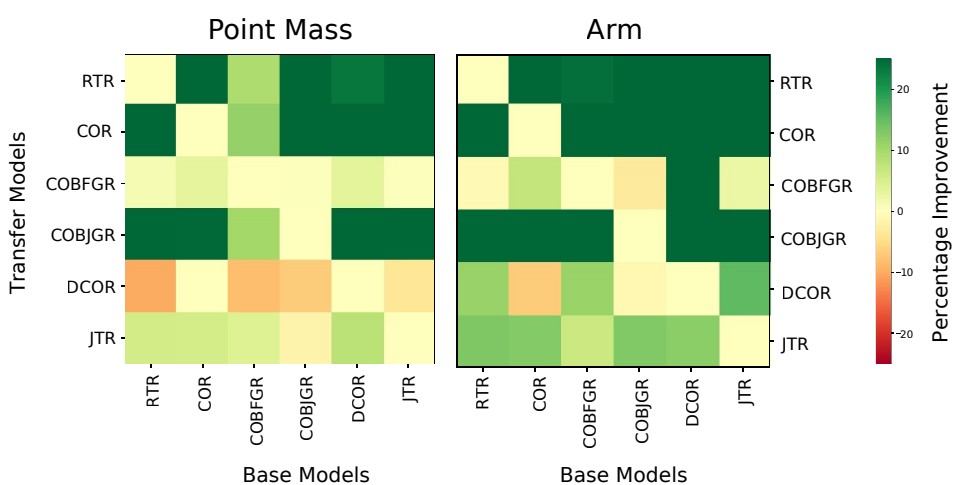

Figure 6: Percentage improvement in AUC of transfer models for Point Mass (left) and Arm (right). Each cell shows a combination of base and transfer tasks, whose performance is then compared to the No-transfer counterpart. The X-axis shows Base Models onto which the transfer models are trained, shown on the Y-axis. (For e.g. the top right cell denotes the percentage improvement of transfer task [RTR on a network previously trained on JTR] from a No-Transfer task [RTR trained from scratch]

Refer to Figure 2 for definition of base, transfer and no-transfer models. Warmer colors (red/yellow) indicate limited or negative transfer, while cooler colors (green) reflect stronger positive transfer. The Arm exhibits more consistent and higher transfer gains across all tasks.

Improvement for the arm model is higher than for the point mass for all task combinations. This trend is present in all task combinations, but it is most pronounced in the intermediate difficulty tasks (COBJGR) where the Arm's AUC improves by 30-35% compared to only 15-20% with the Point Mass (Figure 7 (right)). For the harder tasks (COBFGR and DCOR) the Point Mass shows negligible and even negative transfer, highlighting its limited ability to generalize across temporally constrained tasks. In contrast, the Arm model consistently maintains positive transfer even on more challenging tasks. These gains are all statistically significant, with the exception being COBFGR,

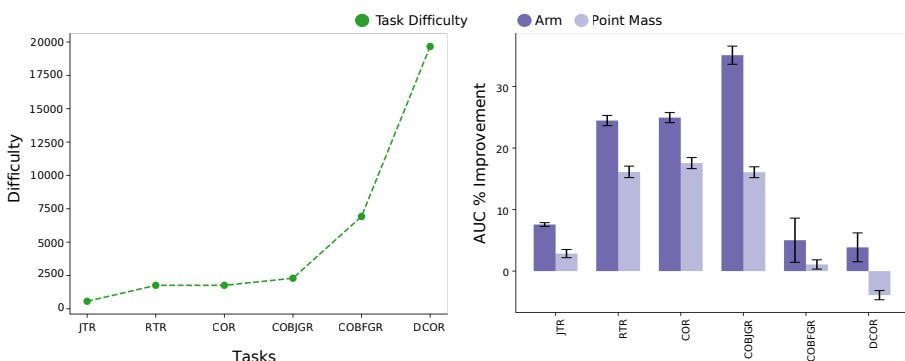

Figure 7: Left: Task difficulty measured in time taken (epochs) to reach stable performance. Right: Comparison between Arm and Point Mass

where performance is statistically indistinguishable (ns) (Figure 7 (right) and Figure 8 Appendix). Thus, averaged across all tasks, the Arm model consistently achieved larger improvements in performance due to transfer learning (Figure 6). Put simply, a complex effector shows greater improvement with transfer learning.

In principle, greater improvement of the arm model due to transfer learning could simply be a consequence of lower No-Transfer performance for the arm model. To address this, we compared the performance of both effector models after transfer learning directly. We quantified the relative performance of the models using the metric $AUC_{Arm:Transfer} - AUC_{PointMass:Transfer}$, that is, the difference in transfer performance across effectors. As shown in Figure 5, this metric is predominantly positive for half of the tasks we examined, in stark contrast to the difference in No-transfer performance, which is exclusively negative (Figure 4). Counterintuitively, this demonstrates that complex effectors not only improve more than simpler effectors through transfer learning, but can also outperform simple effectors after transfer learning for some tasks (Figure 5). Strikingly, this suggests that an increase in effector complexity can actually lead to an improvement in transfer learning efficiency.

Collectively, these observations not only support the hypothesis that embodiment impacts learning, they support the stronger hypothesis that a more complex body can actually facilitate learning.

## 4    CONCLUSION

The role of embodiment in transfer learning is unexplored. Our work advances our understanding by comparing embodied agents with different effectors across a family of two-dimensional reaching tasks. We find that while simpler effectors with orthogonal muscles and no joints are more effective in learning from scratch, more complex bio-mechanically inspired effectors benefit more from transfer learning. These findings suggest that embodiment is not merely a constraint but a variable that can shape an agent's ability to transfer information across diverse tasks.

Future work may extend this framework to non-reaching tasks such as object manipulation or train a policy on more than one transfer task (e.g. having the same policy learn all tasks, combining this study with work from Lazzari & Saxena (2025)). It would also be valuable to explore the role of inductive biases and structured dynamics that could form the theoretical framework to explain our observed results. By re-framing properties of the body as influential components of a learning system, our findings open new directions for investigating embodiment as a resource for skill acquisition and transfer.

## 5 ETHICS STATEMENT

This work does not involve human subjects, personally identifiable information, or sensitive data. The simulation environments (Motornet) and software packages used are publicly available/ open-source and no privacy or security risks are posed. The research is intended for advancing understanding of transfer learning in embodied agents with potential applications to robotics, machine learning and neuroscience communities, with no foreseeable harmful applications. AI/LLMs were only used for minor edits to the writing and structure. The authors acknowledge that we have read and adhered to the ICLR Code of Ethics throughout this study.

## 6 REPRODUCIBILITY STATEMENT

We have taken several steps to ensure reproducibility. Training details are provided in Section 2.1 of the main text. We also include code snippets in the Appendix that thoroughly illustrate the structure of our tasks, effectors, and training. While these snippets are not sufficient to run the full project end-to-end, they document the critical implementation details. Our work additionally builds on an open-source package, Motornet (https://www.motornet.org) referenced in Section 2.4), which allows readers to reproduce the core components. Upon acceptance, we plan to release a GitHub repository containing the full implementation to further support reproducibility.

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

# A APPENDIX

## A.1 DETAILED FIGURES

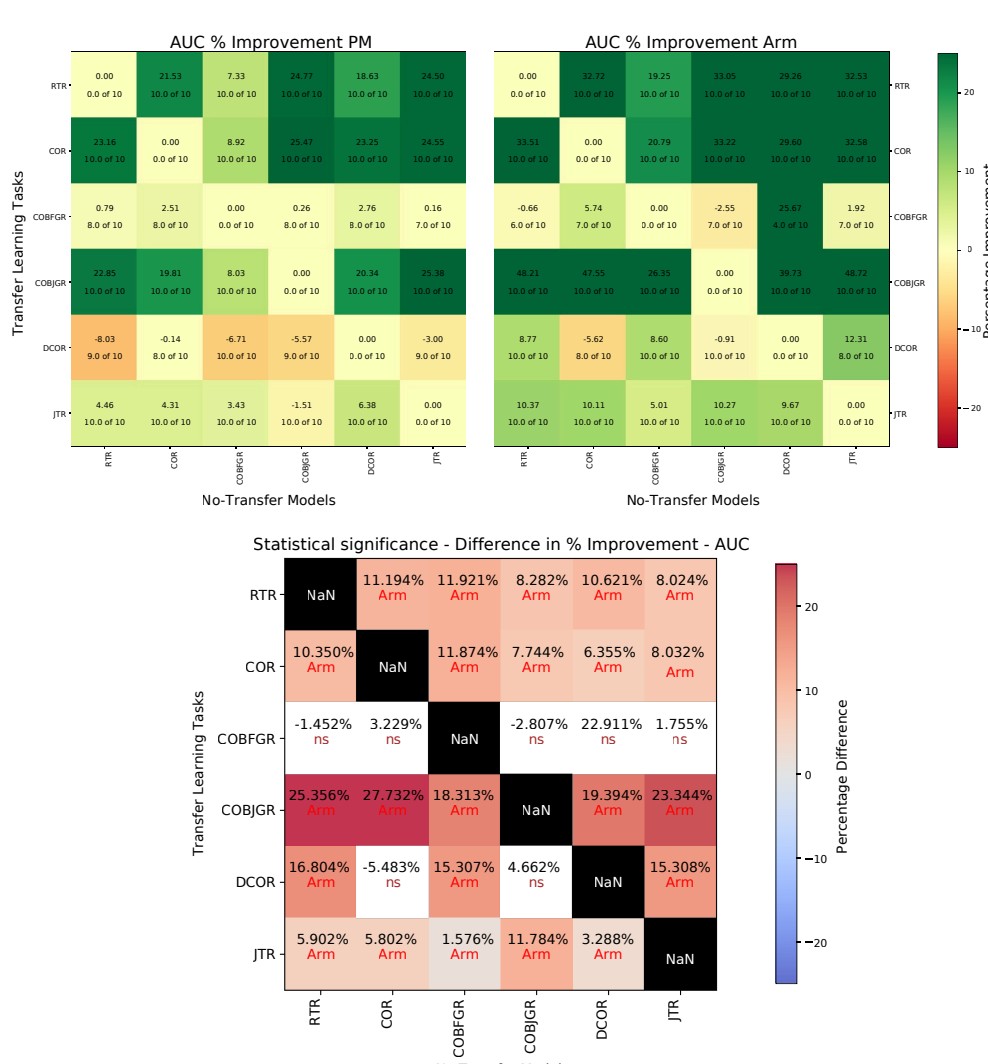

Figure 8: AUC Results

## A.2 CODE FOR TRAINING

Documentation for Motornet: http://motornet.org Documentation for Arm and Point Mass: https://www.motornet.org/tutorials/build-effector.html

Training is a two-stage process involving both from-scratch learning and transfer learning. Models are instantiated as gated recurrent unit (GRU)-based policies with learnable initial hidden states, enabling adaptation to sequential dynamics. Each policy interacts with a differentiable biomechanical effector (either the Point Mass or Arm) within a Motornet environment, where proprioceptive and action signals form the state representation. The reward structure is task-specific, with both dense (distance-based) and structured feedback (e.g., delayed or return-to-center goals) designed to probe temporal and control complexity. Training runs are managed through scripted pipelines that define optimizer settings, learning schedules, and checkpointing, ensuring comparability across task–effector combinations.

Although the training environment provides trial-based rewards, our models are not trained with reinforcement learning. Instead, the entire agent–effector system is differentiable, allowing us to treat the reward signal as a supervised objective function. At each timestep, task-specific performance measures (e.g., distance to target, time in goal zone) are converted into a differentiable loss, and network parameters are updated via standard gradient descent. Thus, training proceeds in a supervised fashion: the policy is optimized directly against continuous task objectives without stochastic policy gradients, value functions, or exploration strategies typically associated with RL. This framing distinguishes our approach from reinforcement learning and situates it more accurately within differentiable control and supervised sequence learning.

```python
import motornet as mn
class Environment(mn.environment.Environment):
    """
    This class modifies the version in Motornet by changing some aspects
        of how the observations,
    noise, and rewards are processed.
    """
    def __init__(
            self,
            effector: ConstrainedGoalReluPointMass |
                ConstrainedGoalRigidTendonArm,
            efferent_noise: float = 0.,
            target_radius: float = 0.,
            noise_type: Literal['additive', 'multiplicative'] | None =
                None,
            action_transform: Callable[[torch.Tensor], torch.Tensor] |
                None = None,
            action_low: float = 0.,
            action_high: float = 1.,
            reward_type: Literal['distance', 'disc'] | None = None,
            **kwargs
    ):
        self.target_radius = target_radius
        self._apply_noise: Callable[[torch.Tensor, torch.Tensor], torch.
            Tensor] = (lambda x, noise: x * (1 + noise)) if noise_type ==
             'multiplicative' else (lambda x, noise: x + noise)
        self.action_transform = action_transform
        # self.reward_type = 'disc' if reward_type is None else
            reward_type
        self.reward_type = 'distance'
        super().__init__(effector, **kwargs)
        self.efferent_noise = [efferent_noise]
        self.effector = effector
        assert isinstance(self.action_space, gym.spaces.Box), f'`
            action_space` is not appropriately initialized.'
        self.action_space = gym.spaces.Box(low=action_low, high=
            action_high, shape=self.action_space.shape, dtype=self.
            action_space.dtype.type)
```

```python
    def get_vision(self) -> torch.Tensor:
        '''
        Provides visual feedback to the neural network about the current
        fingertip position using a normalized coordinate system.
        Called by the inherited update_obs_buffer
        '''
        return 2 * (self.states["fingertip"] - self.effector.xy_goal_min)
            / (self.effector.xy_goal_max - self.effector.xy_goal_min) -
            1

    # def _get_vision(self):
    #     self.dot_width = 0.05
    #     self.num_pixels = 28
    #     # self.pixel_size = 2 / 28
    #     xy = 2 * (self.states["fingertip"] - self.effector.xy_goal_min)
        / (self.effector.xy_goal_max - self.effector.xy_goal_min) - 1
    #     x = y = torch.linspace(-1, 1, self.num_pixels)
    #     x = torch.tile(x, [self.num_pixels, 1])
    #     y = torch.tile(y[:, None], [1, self.num_pixels])
    #     locs = torch.stack([x, y], dim=-1)
    #     return (((locs - xy) / (2 * self.dot_width))**2).sum(dim=-1).
        exp().flatten()  # modolo batches

    def get_proprioception(self) -> torch.Tensor:
        if not hasattr(self, '_mlen_idx'):
            self._mlen_idx = self.muscle.state_name.index('muscle length'
                )
        mlen = self.states["muscle"][:, self._mlen_idx:self._mlen_idx +
            1] / self.muscle.l0_ce
        if not hasattr(self, '_mvel_idx'):
            self._mvel_idx = self.muscle.state_name.index('muscle
                velocity')
        mvel = self.states["muscle"][:, self._mvel_idx:self._mvel_idx +
            1] / self.muscle.vmax
        return torch.cat([mlen, mvel], dim=-1).squeeze(dim=1)

    def _build_spaces(self):
        self.goal_delay = self.vision_delay
        super()._build_spaces()

    # The following properties are useful because the TorchRL libraries
        don't all allow kwarg passing in reset() and step()
    @property
    def step_kwargs(self) -> dict[str, Any]:
        if not hasattr(self, '_step_kwargs'):
            self._step_kwargs = {}
        return self._step_kwargs

    @step_kwargs.setter
    def step_kwargs(self, kwargs: dict[str, Any]) -> None:
        self._step_kwargs = kwargs

    @property
    def reset_kwargs(self) -> dict[str, Any]:
        if not hasattr(self, '_reset_kwargs'):
            self._reset_kwargs = {}
        return self._reset_kwargs

    @reset_kwargs.setter
    def reset_kwargs(self, kwargs: dict[str, Any]) -> None:
        self._reset_kwargs = kwargs

    @property
    def th_random(self):
```

```
            return self.effector.th_random

    # The following properties are a hack that allow for noise sampling
        with pytorch
    @property
    def obs_noise(self) -> torch.Tensor:
        return self._obs_noise

    @obs_noise.setter
    def obs_noise(self, value: Sequence[float]) -> None:
        self._obs_noise = torch.tensor(value, dtype=torch.float32, device
            =self.device)

    @property
    def action_noise(self) -> torch.Tensor:
        return self._action_noise

    @action_noise.setter
    def action_noise(self, value: Sequence[float]) -> None:
        self._action_noise = torch.tensor(value, dtype=torch.float32,
            device=self.device)

    @property
    def vision_noise(self) -> torch.Tensor:
        return self._vision_noise

    @vision_noise.setter
    def vision_noise(self, value: Sequence[float]) -> None:
        self._vision_noise = torch.tensor(value, dtype=torch.float32,
            device=self.device)

    @property
    def efferent_noise(self) -> torch.Tensor:
        return self._efferent_noise

    @efferent_noise.setter
    def efferent_noise(self, value: Sequence[float]) -> None:
        self._efferent_noise = torch.tensor(value, dtype=torch.float32,
            device=self.device)

    @property
    def proprioception_noise(self) -> torch.Tensor:
        return self._proprioception_noise

    @proprioception_noise.setter
    def proprioception_noise(self, value: Sequence[float]) -> None:
        self._proprioception_noise = torch.tensor(value, dtype=torch.
            float32, device=self.device)

    def apply_noise(self, loc, scale: torch.Tensor | Sequence[float] |
        float):
        scale = scale if isinstance(scale, torch.Tensor) else torch.
            tensor(scale, dtype=torch.float32, device=self.device)
        white_noise = torch.randn(size=loc.shape, generator=self.
            th_random, dtype=torch.float32, device=self.device) * scale
        return self._apply_noise(loc, white_noise)

    def get_obs(self, action=None, deterministic: bool = False) -> torch.
        Tensor | np.ndarray:
        '''
        The parent version of this method applies noise twice, once when
        saved to 'obs_buffer' and once
        when retreived from 'obs_buffer'. This version only applies noise
        during retrevial and ignores
        the 'obs_noise' parameter
```

```
138                '''
139            self.update_obs_buffer(action=action)
140            no_op: Callable[[torch.Tensor, torch.Tensor]] = lambda x, y: x
141            f = no_op if deterministic else self.apply_noise
142            obs = (
143                [
144                    self.obs_buffer["goal"][0] if deterministic else self.
                            apply_noise(self.obs_buffer["goal"][0], self.
                            vision_noise),
145                    self.obs_buffer["vision"][0] if deterministic else self.
                            apply_noise(self.obs_buffer["vision"][0], self.
                            vision_noise),
146                    self.obs_buffer["proprioception"][0] if deterministic
                            else self.apply_noise(self.obs_buffer["proprioception
                            "][0], self.proprioception_noise),
147                ] +
148                [
149                    action if deterministic else self.apply_noise(action,
                            self.efferent_noise) for action in self.obs_buffer["
                            action"]
150                ]
151            )
152            obs = torch.cat(obs, dim=-1)
153            return obs if self.differentiable else self.detach(obs)
154
155        def step(  # type: ignore
156            self,
157            action: torch.Tensor | np.ndarray | Sequence[float],
158            **kwargs
159        ):
160            kwargs = self.step_kwargs | kwargs
161            deterministic = kwargs.pop("deterministic", False)
162            self.elapsed += self.dt
163
164            action = action if isinstance(action, torch.Tensor) else torch.
                    tensor(action, dtype=torch.float32, device=self.device)
165            if self.action_transform is not None:
166                action = self.action_transform(action)
167            noisy_action = action if deterministic else self.apply_noise(
                    action, self.action_noise)
168
169            self.effector.step(noisy_action, **kwargs)
170
171            reward = self._get_reward(self.effector.batch_size)
172            obs = self.get_obs(action=action, deterministic=deterministic)
173
174            truncated = torch.zeros(reward.shape, dtype=torch.bool, device=
                    self.device) if self.differentiable else np.zeros(reward.
                    shape, dtype=bool)
175            terminated = ~truncated & (self.elapsed > self.max_ep_duration or
                     bool(np.isclose(self.elapsed, self.max_ep_duration)))
176
177            info = {
178                "states": self._maybe_detach_states(),
179                "action": action if self.differentiable else self.detach(
                        action),
180                "noisy_action": noisy_action if self.differentiable else self
                        .detach(noisy_action),
181                "goal": self.goal if self.differentiable else self.detach(
                        self.goal),
182            }
183
184            return obs, reward, terminated, truncated, info
185
```

```
186     def _get_reward(self, batch_size: int) -> np.ndarray | torch.Tensor:
            # type: ignore
187         error = (self.goal - self.states['fingertip']).norm(dim=-1,
                keepdim=True) / self.target_radius
188         self._at_goal = error < 1
189         if self.reward_type == 'distance':
190             reward = torch.where(self._at_goal, 1 - error**2, 2 - 2 *
                    error) * self.dt
191         else:
192             reward = torch.where(self._at_goal, self.dt, 0.)
193         return reward if self.differentiable else self.detach(reward)
194
195     def reset(self, **kwargs) -> tuple[Any, dict[str, Any]]:
196         return super().reset(**(self.reset_kwargs | kwargs))
```

Listing 1: Training Environment

```
1  class RandomTargetReach(Environment): # % time on target
2    def _get_goal(self, batch_size: int) -> torch.Tensor:
3      return self.joint2cartesian(self.effector.draw_random_uniform_states(
           batch_size)).chunk(2, dim=-1)[0]
```

Listing 2: Random Target Reach

```
1  class CenterOutReach(RandomTargetReach): # % time on target
2      def __init__(self, effector: ConstrainedGoalReluPointMass |
           ConstrainedGoalRigidTendonArm, *args, **kwargs):
3          super().__init__(effector, *args, q_init=effector.center, **
               kwargs)
4
5      @property
6      def cartesian_center(self):
7          if not hasattr(self, '_cartesian_center'):
8              joint_state = torch.cat((self.effector.center, torch.
                   zeros_like(self.effector.center)), dim=-1)
9              self._cartesian_center = self.joint2cartesian(joint_state).
                   chunk(2, dim=-1)[0]
10         return self._cartesian_center
```

Listing 3: Center Out Reach

```
1  class CenterOutAndBackWithJumpingGoalReach(CenterOutReach):
2      def _get_reward(self, batch_size: int):
3          reward = super()._get_reward(batch_size)
4          self.goal = torch.where(self._at_goal, self.cartesian_center,
               self.goal)
5          return reward
```

Listing 4: COBJGR

```
1  class CenterOutAndBackWithFixedGoalReach(CenterOutReach):
2      def __init__(self, effector: ConstrainedGoalReluPointMass |
           ConstrainedGoalRigidTendonArm, *args, **kwargs):
3          super().__init__(effector, *args, **kwargs)
4          self.reward_offset = 0
5          self.rewarded_goal = self.goal
6          self.previous_fingertip = None # Store the previous fingertip
               position
7          self.goal_hold_counter =  None # Counter for how long the goal
               has been held, originally 0
8      def _get_reward(self, batch_size: int):
9          visual_goal = self.goal  # Store the current goal temporarily
10         self.goal = self.rewarded_goal  # Use the rewarded goal for
               reward calculation
```

```python
            current_reward = super()._get_reward(batch_size)
            at_goal = self._at_goal

            reward = current_reward + self.reward_offset

            # Initialize hold counter
            if self.goal_hold_counter is None:
                self.goal_hold_counter = torch.zeros((batch_size,1),device=
                    self.device)

            # Hold at the goal before switching to the return phase
            self.goal_hold_counter = torch.where(at_goal, self.
                goal_hold_counter+1,torch.zeros_like(self.goal_hold_counter))
                    # Increment the counter if at goal, reset otherwise
            hold_steps = 5 # Define how many steps to hold at the goal before
                 switching to the return phase
            self.switch_return = self.goal_hold_counter >= hold_steps #
                Switch to return phase if hold steps are reached

            # Adjust the rewarded goal if the effector reaches the current
                goal & waits enough steps
            self.goal = self.rewarded_goal = torch.where(self.switch_return,
                self.cartesian_center, self.rewarded_goal)
            temp_reward = super()._get_reward(batch_size)  # Calculate reward
                 for the new goal
            self.reward_offset = torch.where(self.switch_return,
                current_reward - temp_reward, 0) + self.reward_offset
            self.goal = visual_goal  # Restore the original current goal
            return reward

    def step(self, *args, **kwargs):
        obs, reward, terminated, truncated, info = super().step(*args, **
            kwargs)
        info['goal'] = self.rewarded_goal if self.differentiable else
            self.detach(self.rewarded_goal)

        # Calculate feedback indicating if the current goal matches the
            rewarded goal
        feedback = (self.rewarded_goal == self.goal).to(dtype=torch.float
            )

        # Taking mean
        feedback = feedback.mean(dim=-1, keepdim=True)  # Convert [
            batch_size, 2] to [batch_size, 1]

        # Concatenate feedback with observations
        obs = torch.cat((obs, feedback), dim=1)

        return obs, reward, terminated, truncated, info

    def reset(self, *args, **kwargs):
        obs, info = super().reset(*args, **kwargs)
        self.rewarded_goal = self.goal
        self.previous_fingertip = None  # Reset previous fingertip
            position
        self.goal_hold_counter = None

        # Concatenate initial feedback (set to 1) with observations
        obs = torch.cat((obs, torch.ones_like(obs[:, :1])), dim=1)

        return obs, info
```

Listing 5: COBFGR

```python
class DelayedCenterOutReach(CenterOutReach):
    def __init__(self, *args, delay: float = 0., **kwargs):
        self.delay = delay
        super().__init__(*args, **kwargs)
    @property
    def waiting(self):
        return self.elapsed < self.delay or bool(np.isclose(self.elapsed,
            self.delay))
    def _get_reward(self, batch_size: int):
        goal = self.goal
        if self.waiting:
            self.goal = self.cartesian_center.expand_as(goal)
        reward = super()._get_reward(batch_size)
        self.goal = goal
        return reward
    def step(self, *args, **kwargs):
        obs, reward, terminated, truncated, info = super().step(*args, **
            kwargs)
        goal = self.cartesian_center.expand_as(self.goal) if self.waiting
            else self.goal
        info['goal'] = goal if self.differentiable else self.detach(goal)
        return obs, reward, terminated, truncated, info
    def reset(self, *args, **kwargs):
        obs, info = super().reset(*args, **kwargs)
        goal = self.cartesian_center.expand_as(self.goal) if self.waiting
            else self.goal
        info['goal'] = goal if self.differentiable else self.detach(goal)
        return obs, info
```

Listing 6: DCOR

```python
class JumpingTargetReach(RandomTargetReach): #num targets hit
    def _get_reward(self, batch_size: int):
        current_reward = super()._get_reward(batch_size) # negative
            distance to the current goal
        reward = current_reward + self.reward_offset
        # save reward data at every timestep
        num_jumps = int(self._at_goal.sum())
        self.num_goals_reached += self._at_goal
        if num_jumps > 0:
            new_goals = self._get_goal(num_jumps)
            idx = self._at_goal.nonzero()[:, :1].expand_as(new_goals)
            self.goal = self.goal.scatter(0, idx, new_goals)
            temp_reward = super()._get_reward(batch_size) # negative
                distance to the new goals
            self.reward_offset = current_reward - temp_reward + self.
                reward_offset
        return reward
```

Listing 7: JTR

```python
import subprocess
from time import sleep
import os
import time
import torch

def get_checkpoint_info(model_path):
    """
    Extracts the checkpoint information from the model file.
    """
    try:
        checkpoint = torch.load(model_path, map_location='cpu')
        epochs_completed = checkpoint.get('epochs_run',0)
```

```python
            return epochs_completed
    except Exception as e:
            print(f"Error reading checkpoint {model_path}:{e}")
            return 0

def launch_training(model_filename, effector_type, task, name, lr, epochs
    , batch_size, hidden_sizes, muscle_weights, hidden_act_weight,
    hidden_act_deriv_weight, debug=False, log_dir="logs11"):
    outstr = f"""#!/bin/sh
#
# Training with model {model_filename} on task {task}
#
#SBATCH --account=X
#SBATCH --job-name={model_filename}_{task}      # The job name.
#SBATCH -c 1                          # The number of cpu cores to use
#SBATCH -t 0-12:00                    # Runtime in D-HH:MM
#SBATCH --mem=32gb
#SBATCH -o {log_dir}/{task}_{name}.out # Create output files for each
    training run
#SBATCH -e {log_dir}/{task}_{name}.err # Create error files for each
    training run
echo Job ID: $SLURM_JOBID

# module load anaconda
source activate embodied_metalearning
export PYTHONUNBUFFERED=TRUE
echo "SLURM job running on: $(hostname)"
echo "Python path: $(which python)"
echo "Conda env: $CONDA_DEFAULT_ENV"
python --version

python retrain_models1.py \\
  --effector-type {effector_type} \\
  --task {task} \\
  --file {model_filename} \\
  --epochs {epochs} \\
  --batch-size {batch_size} \\
  --hidden-sizes {hidden_sizes} \\
  --lr {lr} \\
  --load-file {model_filename}
"""

    script_filename = "tmp.sh"
    with open(script_filename, "w") as text_file:
        text_file.write(outstr)

    subprocess.run(["sbatch", script_filename])

    print(f"Script saved as {script_filename}")

# Base model training epochs
base_model_epochs = {
    'RandomTargetReach': 5000,
    'CenterOutReach': 5000,
    'CenterOutAndBackWithJumpingGoalReach': 5000,
    'CenterOutAndBackWithFixedGoalReach': 35000,
    'DelayedCenterOutReach': 35000,
    'JumpingTargetReach': 5000,
}

task_parameters = {
    'RandomTargetReach': {
        'lr': 1e-3,
        'epochs': 5000,
        'batch_size': 128,
```

```
 75          'hidden_sizes': "32_16",
 76          'muscle-weight': 0,
 77          'hidden-act-weight': 100,
 78          'hidden-act-deriv-weight': 0.005
 79      },
 80      'CenterOutReach': {
 81          'lr': 1e-3,
 82          'epochs': 5000,
 83          'batch_size': 128,
 84          'hidden_sizes': "32_16",
 85          'muscle-weight': 0,
 86          'hidden-act-weight': 100,
 87          'hidden-act-deriv-weight': 0.005
 88      },
 89      'CenterOutAndBackWithJumpingGoalReach': {
 90          'lr': 1e-3,
 91          'epochs': 5000,
 92          'batch_size': 128,
 93          'hidden_sizes': "32_16",
 94          'muscle-weight': 0,
 95          'hidden-act-weight': 100,
 96          'hidden-act-deriv-weight': 0.005
 97      },
 98      'CenterOutAndBackWithFixedGoalReach': {
 99          'lr': 1e-4,
100          'epochs': 35000,
101          'batch_size': 512,
102          'hidden_sizes': "128_16",
103          'muscle-weight': 0,
104          'hidden-act-weight': 100,
105          'hidden-act-deriv-weight': 0.0005
106      },
107      'DelayedCenterOutReach': {
108          'lr': 1e-3,
109          'epochs': 35000,
110          'batch_size': 128,
111          'hidden_sizes': "32_16",
112          'muscle-weight': 0,
113          'hidden-act-weight': 100,
114          'hidden-act-deriv-weight': 0.005
115      },
116      'JumpingTargetReach': {
117          'lr': 1e-3,
118          'epochs': 5000,
119          'batch_size': 128,
120          'hidden_sizes': "32_16",
121          'muscle-weight': 0,
122          'hidden-act-weight': 100,
123          'hidden-act-deriv-weight': 0.005
124      },
125 }
126
127 def main():
128     models = 'leftmodels'  # Directory containing model files
129     # tasks = [
130     #     "RandomTargetReach",
131     #     "CenterOutReach",
132     #     "CenterOutAndBackWithJumpingGoalReach",
133     #     "CenterOutAndBackWithFixedGoalReach",
134     #     "DelayedCenterOutReach",
135     #     "JumpingTargetReach"
136     # ]
137
138     log_dir = "logs11"
139
```

```
        os.makedirs(log_dir, exist_ok=True)

        # List of models
        model_filenames = [f for f in os.listdir(models) if f.endswith('.pt')
            ]
        print(f"Found {len(model_filenames)} models.")

        # Iterate through models and tasks
        for model_filename in model_filenames:
            model_path = os.path.join(models,model_filename)

            #get the number of epochs completed from the model file
            #epochs_completed = get_checkpoint_info(model_path)

            effector_type = "arm" if "arm" in model_filename else "point-mass
                "

            parts = model_filename.split('_over_')
            target_task = parts[0].split('_')[1] # the task to be trained
            name = parts[1].split('.pt')[0] # the name of the base model file
            trained_task = name.split('_')[1] # base task

            # Get the number of epochs the base model was trained on
            base_model_trained_epochs = base_model_epochs.get(trained_task,
                0)

            print(f"Continute training model {model_filename} on task {
                target_task} with effector type {effector_type}")

            params = task_parameters[target_task]

            # Calculate total epochs (base model epochs + new task epochs)
            total_epochs = base_model_trained_epochs + params["epochs"]

            #left_epochs = total_epochs - epochs_completed

            launch_training(
                    model_filename=model_filename,
                    effector_type=effector_type,
                    task=target_task,
                    name=name,
                    lr=params["lr"],
                    epochs= total_epochs,   # Pass total epochs
                    batch_size=params["batch_size"],
                    hidden_sizes=params["hidden_sizes"],
                    muscle_weights=params["muscle-weight"],
                    hidden_act_weight=params["hidden-act-weight"],
                    hidden_act_deriv_weight=params['hidden-act-deriv-weight']
                )

            print(f"Training command sent.")

if __name__ == "__main__":
    main()
```

Listing 8: Transfer Learning Hyperparameters

```
def build_env(seed):
    env = make_motornet_env(differentiable=True, effector_type=
        effector_type, device=device, task=task, target_radius_frac=
        target_radius_frac, efferent_noise=noise_level, vision_noise=
        noise_level, proprioception_noise=noise_level, action_noise=
        noise_level)
    env._set_generator(seed)
    return env
```

```python
def build_actor(obs_dim, action_dim):
    actor = GRUActor(obs_dim, action_dim, hidden_sizes, hidden0_sigma=
        h0_scale, positive_func=positive_func, separate_scale=
        separate_scale, scale_hidden_sizes=hidden_sizes[1:], manual_BG=
        rl_for_BG, device=device, dtype=dtype)
    if distributed:
        actor = DDP(actor)
    return actor

def build_critic(obs_dim):
    critic = GRUCritic(obs_dim, hidden_sizes, hidden0_sigma=h0_scale,
        device=device, dtype=dtype)
    if distributed:
        critic = DDP(critic)
    return critic

# %% [markdown]
# Helper functions for saving and loading checkpoints. Model will be
#     loaded if `load_file` exists. (Note: if command line option `--load-
#     file` is not used then `load_file` will equal `file`.)

# %%
def _load(env, actor, critic, optimizer, scheduler):
    defaults = 0, {'losses': [], 'actor_losses': [], 'critic_losses': [],
        'hidden_unit_losses': [],
                   'muscle_activation_losses': [],'
                       avg_hidden_activity_derivative': [], 'tar_frac':
                       []}
    if load_file is None:
        return defaults
    try:
        snapshot = th.load(load_file)
    except FileNotFoundError:
        return defaults
    print(f'Restoring model from checkpoint in {load_file}.', file = sys.
        stdout)

    env.load_state_dict(snapshot['env_dict'])
    (actor.module if distributed else actor).load_state_dict(snapshot['
        actor_dict'])
    (critic.module if distributed else critic).load_state_dict(snapshot['
        critic_dict'])

    optimizer.load_state_dict(snapshot['optimizer_dict'])
    scheduler.load_state_dict(snapshot['scheduler_dict'])

    if len(snapshot['torch_states']) != world_size:
        warn(f'Cannot restore the random state because number saved
            states = {len(snapshot["torch_states"])} and world_size = {
            world_size}.', category=RuntimeWarning)
    else:
        torch.set_rng_state(snapshot['torch_states'][rank])
        env.np_random.bit_generator.state = snapshot['numpy_env_states'][
            rank]
    if rank == 0:
        return snapshot['epochs_run'], snapshot['stats']
    else:
        return snapshot['epochs_run'], defaults[1]

def load_actor_weights(actor, snapshot):
    actor_dict = snapshot['actor_dict']
    model_dict = actor.state_dict()
```

```
55      # Filter out mismatched layers
56      filtered_actor_dict = {k: v for k, v in actor_dict.items() if k in
            model_dict and v.size() == model_dict[k].size()}
57
58      # Load the filtered weights
59      (actor.module if distributed else actor).load_state_dict(
            filtered_actor_dict, strict=False)
60      print(f"Actor weights partially loaded. {len(filtered_actor_dict)}
            layers were loaded successfully.")
61
62  def load_critic_weights(critic, snapshot):
63      critic_dict = snapshot['critic_dict']
64      model_dict = critic.state_dict()
65
66      # Filter out mismatched layers
67      filtered_critic_dict = {k: v for k, v in critic_dict.items() if k in
            model_dict and v.size() == model_dict[k].size()}
68
69      # Load the filtered weights
70      (critic.module if distributed else critic).load_state_dict(
            filtered_critic_dict, strict=False)
71      print(f"Critic weights partially loaded. {len(filtered_critic_dict)}
            layers were loaded successfully.")
72
73  def _load_weights(env, actor, critic, optimizer, scheduler):
74      print('I am inside loading weights', file=sys.stdout)
75      defaults = 0, {'losses': [], 'actor_losses': [], 'tar_frac': []}
76      if load_file is None:
77          print('Loading defaults', file=sys.stdout)
78          return defaults
79
80      load_file_path = os.path.join("leftmodels", load_file)
81      print(f'Loading file from path: {load_file_path}', file=sys.stdout)
82
83      if not os.path.exists(load_file_path):
84          print(f'File {load_file_path} does not exist.', file=sys.stdout)
85          return defaults
86
87      try:
88          snapshot = torch.load(load_file_path)
89          print('Restoring model weights from checkpoint', file=sys.stdout)
90      except Exception as e:
91          print(f'Error loading file: {e}', file=sys.stdout)
92          return defaults
93
94      # Load actor weights (with filtering)
95      try:
96          load_actor_weights(actor, snapshot)
97      except Exception as e:
98          print(f'Error loading Actor weights completely: {e}', file=sys.
                stdout)
99
100     # Load critic weights (with filtering)
101     try:
102         load_critic_weights(critic, snapshot)
103     except Exception as e:
104         print(f'Error loading Critic weights completely: {e}', file=sys.
                stdout)
105
106     return snapshot['epochs_run'], snapshot['stats']
107
108
109 def _save(env, actor, critic, optimizer, scheduler, stats, run,
        current_epoch=None): #Command to save .pt file
110     foldername ='continue_models3'
```

```python
    #if folder does not exist, create it
    if not os.path.exists(foldername):
        os.makedirs(foldername)

    #filename = os.path.join(foldername, f'{run}_{task}_over_{file}')
    filename = os.path.join(foldername, f'{file}')
    torch_state = torch.get_rng_state()
    numpy_env_state = env.np_random.bit_generator.state
    if distributed:
        torch_states = [torch.zeros_like(torch_state) for _ in range(
            world_size)]
        torch.distributed.all_gather(torch_states, torch_state)
        numpy_env_states = [None] * world_size
        torch.distributed.all_gather_object(numpy_env_states,
            numpy_env_state)
    else:
        torch_states = [torch_state]
        numpy_env_states = [numpy_env_state]
    if rank == 0:
        snapshot = dict(
            env_dict=env.state_dict(),
            actor_dict=(actor.module if distributed else actor).
                state_dict(),
            critic_dict=(critic.module if distributed else critic).
                state_dict(),
            optimizer_dict=optimizer.state_dict(),
            scheduler_dict=scheduler.state_dict(),
            #epochs_run=len(stats['losses']),
            epochs_run=current_epoch + 1 if current_epoch is not None
                else len(stats['losses']),
            stats=stats,
            torch_states=torch_states,
            numpy_env_states=numpy_env_states,
        )
        th.save(snapshot, filename)

# %% [markdown]
# This function updates the basal ganglia weights using the advantages
#    when the command line option '--rl-for-BG' is used. If the command
#    line option '--check-grad' is also used, then automatic
#    differentiation is also used to make sure the gradient calculations
#    are correct. Gradients are clipped to be less than `max_norm` when
#    that argument is not `None`.

# %%
def update_BG_weights(adv, ac, loc, scale, ctx, str, actor, lr=1e-3,
    max_norm=None, detach=False):
    for x in (ac, loc, scale, ctx, str):
        assert adv.shape == x[..., :1].shape
    if detach:
        adv = adv.detach()
        ac = ac.detach()
        loc = loc.detach()
        scale = scale.detach()
        ctx = ctx.detach()
        str = str.detach()

    a = ac
    c = ctx
    s = str
    d   = torch.where(s > 0, 1., 0.)
        = loc
        = scale

    actor = actor.module if distributed else actor
```

```python
        cs = actor.bg.corticostriatal[0]
        s  = actor.bg.loc
        if not separate_scale:
            s  = actor.bg.scale[0]

        def apply_adv(w, two_d=False):
            tmp = ((adv[..., None] if two_d else adv) * w).mean(dim=(0, 1))
            if distributed:
                dist.all_reduce(tmp)
            return tmp / world_size

        z = (a -   ) /
        grads = {'cs': {}, 's ': {}}
        tmp_ = z /
        grads['s ']['bias'] = apply_adv(tmp_ )
        grads['s ']['weight'] = apply_adv(tmp_ .unsqueeze(-1) * s.unsqueeze
            (-2), two_d=True)
        tmp_s = tmp_ @ s .weight
        if not separate_scale:
            grads['s '] = {}
            d _over_ = 1 if positive_func == 'exp' else -(-  ).expm1() /

            tmp_ = -d _over_ * (1 - z**2)
            grads['s ']['bias'] = apply_adv(tmp_ )
            grads['s ']['weight'] = apply_adv(tmp_ .unsqueeze(-1) * s.
                unsqueeze(-2), two_d=True)
            tmp_s = tmp_s + tmp_ @ s .weight
        tmp_s = d  * tmp_s
        grads['cs']['bias'] = apply_adv(tmp_s)
        grads['cs']['weight'] = apply_adv(tmp_s.unsqueeze(-1) * c.unsqueeze
            (-2), two_d=True)

        if check_grad:
            assert not distributed, 'Gradient checking currently only works
                for single process training.'
            _params = {}
            sync = lambda source, target: target.load_state_dict(source.
                state_dict())

            _cs = nn.Linear(cs.in_features, cs.out_features, device=device,
                dtype=dtype)
            _s  = nn.Linear(s .in_features, s .out_features, device=device
                , dtype=dtype)
            sync(cs, _cs)
            sync(s , _s )
            if distributed:
                _cs, _s  = DDP(_cs), DDP(_s )
            _params['cs'] = dict(weight=(_cs.module if distributed else _cs).
                weight, bias=(_cs.module if distributed else _cs).bias)
            _params['s '] = dict(weight=(_s .module if distributed else
                _s ).weight, bias=(_s .module if distributed else _s ).
                bias)

            if not separate_scale:
                _s  = nn.Linear(s .in_features, s .out_features, device=
                    device, dtype=dtype)
                sync(s , _s )
                if distributed:
                    _s  = DDP(_s )
                _params['s '] = dict(weight=(_s .module if distributed else
                    _s ).weight, bias=(_s .module if distributed else _s
                    ).bias)

            _c = c.detach()
            _a = a.detach()
```

```python
            _   = actor.bg.corticostriatal[1]
            _s = _   (_cs(_c))
            _   = _s (_s)
            if not separate_scale:
                _   = actor.bg.scale[1]
                _   = _   (_s (_s))
            else:
                _   = _   .detach()

            _pi = th.distributions.Normal(_   , _   )
            _pi = th.distributions.Independent(_pi, 1)
            _lp = _pi.log_prob(_a)
            _adv = adv.detach()
            (_adv[..., 0] * _lp).mean().backward()

            if rank == 0:
                for layer in grads:
                    for type in ('weight', 'bias'):
                        allclose = torch.allclose(_params[layer][type].grad,
                            grads[layer][type], rtol=1e-4, atol=1e-7)
                        if not allclose:
                            print(f'Gradients_mismatch_for_{layer}.{type}._
                                Max_error:_{((_params[layer][type].grad_-_
                                grads[layer][type]).abs()_/_(1e-3_+_grads[
                                layer][type].abs())).max():.3e}', file = sys.
                                stdout)

    _norms = []
    for layer in grads:
        for type in ('weight', 'bias'):
            _norms.append(grads[layer][type].norm())
    norm = torch.stack(_norms).norm()
    info_str = ('Clipping_' if max_norm is not None and max_norm < norm
        else '') + f'BG_grad_{norm:.0e},_'
    if max_norm is not None and max_norm < norm:
        for layer in grads:
            for type in ('weight', 'bias'):
                grads[layer][type] = grads[layer][type] * max_norm / norm

    actor.bg.corticostriatal[0].update(lr * grads['cs']['weight'], lr *
        grads['cs']['bias'])
    actor.bg.loc.update(lr * grads[' s ']['weight'], lr * grads[' s ']['
        bias'])
    if not separate_scale:
        actor.bg.scale[0].update(lr * grads[' s ']['weight'], lr * grads[
            ' s ']['bias'])

    return info_str + f"BG_lr:_{lr:.0e},_"

# %% [markdown]
# The function computes the generalized advantage estimate. In standard
#   RL, this uses `rewards` from an environment rollout and `values`
#   computed by the critic. This function alternatively accepts `deltas`,
#    which are analogous to the TD0 values in standard RL. In our case,
#   we will try to learn `deltas` directly rather than compute them from
#   rewards and values.

# %%
def gae(adv_mat, values=None, rewards=None, deltas=None):
    if deltas is None:
        deltas = rewards + gamma * values[:, 1:] - values[:, :-1]
    adv = (adv_mat * deltas).sum(-2)[..., None]
    vtar = adv + values[:, :-1] if values is not None else None
    return adv, vtar
```

```python
# %% [markdown]
# The find_unique_goals function find the number of goals that appeared
#     for each batch, where performance is measured as the fraction of the
#     number of goals reached over the number of goals that appeared. This
#     function is needed when training on the task "JumpingTargetReach".
#     For this task, tar_frac does not accurately assess performance, as
#     the effector could spend a very small fraction of time at each goal,
#     but have perfect performance.

# %%
def find_unique_goals(tg):
    # Initialize an empty tensor to store the counts
    num_unique_pairs_per_batch = torch.zeros((tg.size(0), 1), dtype=torch
        .int32)

    # Iterate over each batch
    for i in range(tg.size(0)):
        reshaped_batch = tg[i].view(-1, tg.size(-1))

        # Find and count unique goals
        unique_rows = torch.unique(reshaped_batch, dim=0)
        num_unique_pairs = unique_rows.size(0)
        num_unique_pairs_per_batch[i] = num_unique_pairs

    return num_unique_pairs_per_batch

# %% [markdown]
# This function generates `batch_size` parallel rollouts of the
#     environment using the current policy of `actor`.

# %%
def rollout(env, actor, batch_size):
    o, info = env.reset(options={"batch_size": batch_size})
    done = False
    h = None
    is_init = th.ones(batch_size, 1, dtype=bool, device=device)

    ob = [o]
    xy = [info["states"]["fingertip"]]
    joint_states = [info["states"]["joint"]]
    tg = [info["goal"]]
    noisy_action = []
    ac = []
    lp = []
    rw = []
    loc = []
    scale = []
    ctx = []
    str = []

    while not done:  # will run until `max_ep_duration` is reached
        pi,   ,   , striatum, cortex, h = actor((o.to(dtype), is_init, h)
            )
        a = pi.rsample()
        # logp = pi.log_prob(a)

        # compute forces acting on fingertip position
        # cartesian_fingertip_pos = env.effector.joint2cartesian(info["
            states"]["fingertip"])
        # force_vector = force_field_selector(cartesian_fingertip_pos,
            force_field_type, force_field_strength)

        o, r, terminated, truncated, info = env.step(action=a.to(torch.
            float32), endpoint_load=th.tensor(force_field).unsqueeze(0))
        done = terminated.any() | truncated.any()
```

```python
            ob.append(o)
            xy.append(info["states"]["fingertip"])  # trajectories
            joint_states.append(info["states"]["joint"]) # joint states
            tg.append(info["goal"])  # targets
            noisy_action.append(info["noisy_action"])
            ac.append(a)
            # lp.append(logp)
            rw.append(r)
            loc.append(  )
            scale.append(  )
            ctx.append(cortex)
            str.append(striatum)

        ob = th.stack(ob, dim=1)
        xy = th.stack(xy, dim=1)
        joint_states = th.stack(joint_states, dim=1)
        tg = th.stack(tg, dim=1)
        noisy_action = th.stack(noisy_action, dim=1)
        ac = th.stack(ac, dim=1)
        # lp = th.stack(lp, dim=1)
        rw = th.stack(rw, dim=1)
        loc = th.stack(loc, dim=1)
        scale = th.stack(scale, dim=1)
        ctx = th.stack(ctx, dim=1)
        str = th.stack(str, dim=1)
        return ob, xy, joint_states, tg, noisy_action, ac, lp, rw, loc, scale
            , ctx, str

# %% [markdown]
# This function runs and trains the model. Model training can take
    multiple forms according to the command line options. The default is
    supervised learning with no RL.

# %%
def train_model():
    th.manual_seed(torch_seed + rank)
    bs = batch_size // world_size
    if world_size > 1:
        print(f'Worker {rank} with batch_size {bs}', file = sys.stdout)

    env = build_env(np_seed + rank)
    actor = build_actor(env.observation_space.shape[0], env.action_space.
        shape[0])
    critic = build_critic(env.observation_space.shape[0])
    optimizer = th.optim.AdamW((*actor.parameters(), *critic.parameters()
        ), lr=lr)

    scheduler = th.optim.lr_scheduler.CosineAnnealingLR(optimizer, epochs
        , eta_min=lr if lr_min is None else lr_min)
    if exp_state == 'train':
        epochs_run, stats = _load(env, actor, critic, optimizer,
            scheduler)
    else:
        epochs_run, stats = _load_weights(env, actor, critic, optimizer,
            scheduler)
        print('epochs_run is ', epochs_run, file = sys.stdout)
        print('Epoch length is', len(stats['losses']),file = sys.stdout)

    # if exp_state == 'train':
    #     epochs_run, stats = _load(env, actor, critic,None,None)
    # else:
    #     epochs_run, stats = _load_weights(env, actor, critic, None,None
        )
```

```
376     #       print('epochs run is ', epochs_run, file = sys.stdout)
377
378     # remaining_epochs = epochs - epochs_run
379     # scheduler = th.optim.lr_scheduler.CosineAnnealingLR(optimizer,
            remaining_epochs, eta_min=lr if lr_min is None else lr_min)
380
381     # # If we have a checkpoint, load the scheduler state
382     # if exp_state == 'train' and epochs_run > 0:
383     #     try:
384     #         snapshot = th.load(load_file)
385     #         scheduler.load_state_dict(snapshot['scheduler_dict'])
386     #     except:
387     #         print("Could not load scheduler state, using new scheduler
            ")
388
389     adv_mat = th.tensor(
390         circulant((gamma * lmbda)**np.arange(int(round(env.
                max_ep_duration / env.dt)))),
391         dtype=dtype,
392         device=device
393     ).tril()
394     print('before gt-trl', file = sys.stdout)
395     get_rl_lr = lambda batch: rl_lr if rl_lr_min is None else (rl_lr -
            rl_lr_min) / 2 * (1 + np.cos(batch * np.pi / epochs)) + rl_lr_min
396     print('About to enter loop', file = sys.stdout)
397     for batch in range(epochs_run-1, epochs):
398
399         optimizer.zero_grad()
400         info_str = ''
401
402         # this is pretty ugly code...
403         ob, xy, joint_states, tg, noisy_action, ac, lp, rw, loc, scale,
                ctx, str = rollout(env, actor, bs)
404
405         is_init = th.ones(bs, 1, 1, dtype=bool, device=device)
406         vs, _ = critic((ob.to(dtype), is_init, None))
407         adv, vtar = gae(adv_mat, values=vs, rewards=rw)
408
409         if rl_for_BG:
410             if not auto_grad_through_rl:
411                 info_str += update_BG_weights(adv, ac, loc, scale, ctx,
                        str, actor, lr=get_rl_lr(batch), max_norm=
                        max_grad_norm, detach=True)
412             else:
413                 info_str += update_BG_weights(adv, ac, loc, scale, ctx,
                        str, actor, lr=get_rl_lr(batch), max_norm=
                        max_grad_norm)
414                 ob, xy, tg, _, _, rw, *_ = rollout(env, actor, bs)
415                 actor.detach()
416                 vs, _ = critic((ob.to(dtype), is_init, None))
417                 _, vtar = gae(adv_mat, values=vs, rewards=rw)
418
419         # Loss function computation with hidden unit + muscle activity
                regularization
420         actor_loss = -rw.sum(dim=1).mean()
421         # critic_loss = smooth_l1_loss(vs[:, :-1], vtar.detach())
422
423         # hidden unit terms
424         avg_hidden_activity  = ((ctx**2).sum(dim=-1) / ctx.shape[-1]).
                mean()
425         hidden_unit_loss = avg_hidden_activity
426         aug_ctx = torch.cat((actor.gru.hidden0.expand(bs, 1, -1), ctx),
                dim=1)
427         avg_hidden_activity_derivative = ((((aug_ctx[:, 1:] - aug_ctx[:,
                :-1]) / env.dt)**2)/ctx.shape[-1]).sum(dim=-1)
```

```python
            # hidden_unit_loss = hidden_act_deriv_weight *
                avg_hidden_activity_derivative).sum(dim=1).mean()
            avg_hidden_activity_derivative_loss = (
                avg_hidden_activity_derivative).sum(dim=1).mean()

            # muscle activation terms
            normed_max_iso_force = (env.effector.muscle.max_iso_force / (
                torch.sqrt(torch.sum((env.effector.muscle.max_iso_force**2)))
                )).squeeze(1)
            muscle_activation_loss = (torch.matmul(noisy_action,
                normed_max_iso_force.T)**2).squeeze(-1).sum(dim=1).mean()

            # MotorNet Version - L2-norm is squared? (now denonminator is an
                inner product)
            # normed_max_iso_force = (env.effector.muscle.max_iso_force / (
                torch.sum((env.effector.muscle.max_iso_force**2)))).squeeze
                (1)
            # hidden_act_weight*hidden_act_deriv_weight

            # NEW LOSS
            loss = actor_loss + hidden_act_weight*hidden_unit_loss +
                hidden_act_deriv_weight*avg_hidden_activity_derivative_loss +
                 muscle_weight*muscle_activation_loss

            # backward pass & update weights
            loss.backward()
            norm = nn.utils.clip_grad_norm_((*actor.parameters(), *critic.
                parameters()), max_norm=max_grad_norm)   # important!
            info_str += ('Clipping g' if max_grad_norm < norm else 'G') + f'
                rad_{norm:.0e}, '
            optimizer.step()
            scheduler.step()

            if rank == 0:
                if task == "JumpingTargetReach":
                    avg_num_unique_goals = find_unique_goals(tg).to(dtype).
                        mean().item()
                    num_goals_reached = env.num_goals_reached.to(dtype).mean
                        ().item()
                    tar_frac = num_goals_reached/avg_num_unique_goals
                else:
                    tar_frac = ((xy - tg).norm(dim=-1) < env.target_radius).
                        to(dtype).mean().item()
                stats['losses'].append(loss.item())
                stats['actor_losses'].append(actor_loss.item())
                # stats['critic_losses'].append(critic_loss.item())
                stats['hidden_unit_losses'].append(hidden_unit_loss.item()) #
                     add in later
                stats['muscle_activation_losses'].append(
                    muscle_activation_loss.item())
                stats['avg_hidden_activity_derivative'].append(
                    avg_hidden_activity_derivative_loss.item())
                stats['tar_frac'].append(tar_frac)

                # Save checkpoint every 'save_every' epochs
                if batch % SAVE_EVERY_N_EPOCHS == 0:
                    _save(env, actor, critic, optimizer, scheduler, stats,
                        run)
                    print(f'Saved model at batch {batch})', file=sys.stdout)

                if batch % print_every == 0:
                    print('I am in the batch loop epochs run is ', epochs_run
                        , file = sys.stdout)
                    print('but epochs is ', epochs, file = sys.stdout)
```

```python
                    info_str = (
                        f"Batch_{batch:5d}/{epochs:5d},_" +
                        f"tot_loss:_{stats['losses'][-1]:.1e},_" +
                        f"a_loss:_{stats['actor_losses'][-1]:.1e},_" +
                        # f"v loss: {stats['critic_losses'][-1]:.1e}, " +
                        f"h-u_loss:_{stats['hidden_unit_losses'][-1]:.1e},_"
                            +
                        f"ma_loss:_{stats['muscle_activation_losses'][-1]:.1e
                            },_" +
                        f"h-u_derivative:_{stats[
                            'avg_hidden_activity_derivative'][-1]:.1e},_" +
                        f"goal_time:_{int(round(stats['tar_frac'][-1]_*_100))
                            :2d}%,_" +
                        info_str +
                        f"lr:_{optimizer.param_groups[0]['lr']:.0e}"
                    )
                    if 'cur' in locals():
                        info_str += f',_batch_time:_{(time()_-_cur)_/_
                            print_every:.2f}_s'
                    cur = time()
                    print(info_str, file = sys.stdout)
            # if file is not None and batch % save_every == 0:
                # _save(env, actor, critic, optimizer, scheduler, stats)
        return env, actor, critic, optimizer, scheduler, stats

# %% [markdown]
# ## Main program
#
# Setup distributed computing if using.

# %%
distributed = "LOCAL_RANK" in os.environ and "WORLD_SIZE" in os.environ
world_size = int(os.environ["WORLD_SIZE"]) if distributed else 1
rank = int(os.environ["LOCAL_RANK"]) if distributed else 0

# %% [markdown]
# Extract the command line arguments and run the model training loop.

# %%
print('Parser_begins', file = sys.stdout)
parser = argparse.ArgumentParser(prog='random-reach-trainer', description
    ='Trains_either_a_point_mass_with_relu_muscles_or_a_2d_arm_with_hill_
    muscles_(specially_`RigidTendonHillMuscleThelen`_muscles)_on_the_
    reach_to_random_target_task')

parser.add_argument('--task', type=str, default='RandomTargetReach', help
    ='task_name_(default:_%(default)s)')
parser.add_argument('--exp_state', type=str, default='test', help='state_
    of_experiment:_training_or_testing_(default:_%(default)s)')
parser.add_argument('--epochs', type=int, default=3000, help='number_of_
    training_epochs_(default:_%(default)s)')
parser.add_argument('--batch-size', type=int, default=32, help='
    enviroment_rollouts_per_epoch_(default:_%(default)s)')
parser.add_argument('--effector-type', type=str, default='arm', choices=[
    'arm', 'point-mass'], help='type_of_effector_(default:_%(default)s;_
    options:_%(choices)s)')
parser.add_argument('--target-radius-frac', type=float, default=0.05,
    help='tolerance_allowed_around_target_location_as_fraction_of_world_
    size_(default:_%(default)s)')
parser.add_argument('--noise-level', type=float, default=0.02, help='
    noise_value_that_is_used_for_efferent,_visual,_proprioceptive,_and_
    action_noise_(default:_%(default)s)')
parser.add_argument('--force-field', type=float, nargs=2, default=[0, 0],
    help='constants_that_determine_the_strength_of_the_force_applied_in_
    SingleConstantForceFieldHold_task_(default:_%(default)s)')
```

```
parser.add_argument('--hidden-sizes', type=int, nargs=2, default=[32,
    16], help='size of the "cortex" and "striatum" in the model BG (
    default: %(default)s)')
parser.add_argument('--h0-scale', type=float, default=0., help='scale
    between 0 (inclusive) and 1 (exclusive) of the initial recurrent
    activity of the actor and critic GRUs at the beginning of training (
    default: %(default)s)')
parser.add_argument('--positive-func', type=str, default='exp', choices=[
    'exp', 'softplus'], help='function used to ensure positivity of the
    scale of the policy (default: %(default)s; options: %(choices)s)')
parser.add_argument('--critic-weight', type=float, default=1., help='
    weight of critic loss in the total loss (default: %(default)s)')
parser.add_argument('--muscle-weight', type=float, default=0.0, help='
    weight of muscle activation loss in the total loss (default: %(
    default)s)')
parser.add_argument('--hidden-act-weight', type=float, default=1000, help
    ='weight of hidden unit activity loss in the total loss (default: %(
    default)s)')
parser.add_argument('--hidden-act-deriv-weight', type=float, default=0.0,
     help='weight of derivative of hidden unit activity in the hidden
    unit activity loss (default: %(default)s)')
parser.add_argument('--lr', type=float, default=1e-3, help='initial
    learning rate (default: %(default)s)')
parser.add_argument('--lr-min', type=float, help='minimum learning rate
    using cosine annealing (default: no annealing)')
parser.add_argument('--max-grad-norm', type=float, default=1., help='
    Clipping will be applied if gradient norm exceeds this value (default
    : %(default)s)')
parser.add_argument('--rl-for-BG', action='store_true', help='train basal
     ganglia weights with RL')
parser.add_argument('--gamma', type=float, default=0.99, help='reward
    discount rate (default: %(default)s)')
parser.add_argument('--lambda', type=float, default=0.95, help='
    Generalized advantage estimator parameter (default: %(default)s)')
parser.add_argument('--separate-scale', action='store_true', help='when
    training with RL, exclude the pathway which computes the scale of the
     policy')
parser.add_argument('--check-grad', action='store_true', help="check the
    manually computed policy gradient for the basal ganglia weights with
    pytorch's auto grad")
parser.add_argument('--auto-grad-through-rl', action='store_true', help='
    differentiate through RL dynamics (this results in two rollouts per
    epoch)')
parser.add_argument('--rl-lr', type=float, default=1e-3, help='learning
    rate for RL (default: %(default)s)')
parser.add_argument('--rl-lr-min', type=float, help='minimum RL learning
    rate using cosine annealing (default: no annealing)')
parser.add_argument('--print-every', type=int, default=100, help='epochs
    between information string prints (default: %(default)s)')
parser.add_argument('--save-every', type=int, default=100, help='epochs
    between model checkpoint saves (default: %(default)s)')
parser.add_argument('--file', type=str, help='filename for loading and
    saving checkpoints')
parser.add_argument('--load-file', type=str, help='seperate file for
    loading checkpoints (default: same as file)')
parser.add_argument('--seed', type=float, default=0, help='random seed (
    default: %(default)s)')
parser.add_argument('--dtype', type=str, default='float32', choices=['
    float32', 'float64'], help='floating point type (WARNING: Motornet
    converts to float32 so this only applies to actor and critic networks
    ; default: %(default)s; options: %(choices)s)')
parser.add_argument('--device', type=str, default='cpu', choices=['cpu',
    'mps', 'cuda'], help='currently tested only on "cpu", try "mps" and "
    cuda" at your own risk (default: %(default)s; options: %(choices)s)')
    ;
```

```
543  print('parser_ends', file = sys.stdout)
544
545  # %% [markdown]
546  # The default behavior is to load and save from the filename given by the
     #     '--file' option. The '--load-file' option
547  # overrides this behavior such that loading and saving use different
     #     files.
548  #
549  # If you load from a file and the number of epochs requested via the '--
     #     epochs' option is less than or equal to the number in the file,
550  # then the model from the file is loaded but no trained occurs. This is
     #     useful for plotting after training at the
551  # command line.
552  #
553  # If you load from a file and the number of epochs requested is greater
     #     than the number in the file, training will
554  # resume from the loaded checkpoint. In this case you will need to make
     #     sure that all the other options are the
555  # same as during the prior training.
556  #
557  # 'args1', 'args2', 'args3', and 'args4' below correspond to the
     #     following examples.
558  #
559  # Example 1: train from scratch with supervised learning and cosine
     #     annealing.
560  #
561  # Example 2: load saved model which was trained for 3000 epochs with
     #     supervised learning.
562  #
563  # Example 3: continue training from checkpoint after training was aborted
     #      partway through. Note that after training completes, 'completed.pt'
     #     will be the same as 'supervised.pt'. You can check this in the
     #     terminal with the following command: 'python check-model-equivalence.
     #     py supervised.pt completed.pt'.
564  #
565  # Example 4: load saved model which was trained for 3000 epochs with RL
     #     for the basal ganglia. The file 'rl-for-BG.pt' was originally created
     #      from the terminal using a distributed 8 CPU core training scheme: '
     #     $CONDA_PREFIX/bin/torchrun --standalone --nnodes=1 --nproc-per-node=8
     #      random-reach-task.py --epochs 3000 --batch-size 1024 --file rl-for-
     #     BG.pt --lr 1e-3 --lr-min 1e-5 --rl-for-BG --rl-lr .1 --rl-lr-min
     #     .001'.
566  #
567  # %%
568  if is_notebook():
569
570      args1 = [
571          '--task', 'RandomTargetReach',
572          # '--file', 'RandomTargetReach_012_pointmass.pt',
573          '--file', 'RandomTargetReach_arm.pt',
574          '--epochs', '3000',
575          '--batch-size', '128',
576          # '--effector-type', 'point-mass',
577          '--effector-type', 'arm',
578          '--hidden-sizes', '32', '16',
579          '--lr', '1e-3',
580          '--muscle-weight', '0.0',
581          '--hidden-act-weight', '1000.0',
582          '--hidden-act-deriv-weight', '0.0',
583          '--load-file', 'none'
584
585      ]
586
587      args2 = [
588          '--task', 'CenterOutReach',
```

```
589        # '--file', 'CenterOutReach_pointmass.pt',
590        '--file', 'CenterOutReach_arm.pt',
591        '--epochs', '3000',
592        '--batch-size', '128',
593        # '--effector-type', 'point-mass',
594        '--effector-type', 'arm',
595        '--hidden-sizes', '32', '16',
596        '--lr', '1e-3',
597        '--muscle-weight', '0.0',
598        '--hidden-act-weight', '1000.0',
599        '--hidden-act-deriv-weight', '0.0',
600        '--load-file', 'none',
601    ]
602
603    args3 = [
604        '--task', 'CenterOutAndBackWithJumpingGoalReach',
605        # '--file', 'CenterOutAndBackWithJumpingGoalReach_pointmass.pt',
606        '--file', 'CenterOutAndBackWithJumpingGoalReach_arm.pt',
607        '--epochs', '3000',
608        '--batch-size', '128',
609        # '--effector-type', 'point-mass',
610        '--effector-type', 'arm',
611        '--hidden-sizes', '32', '16',
612        '--lr', '1e-3',
613        '--muscle-weight', '0.0',
614        '--hidden-act-weight', '1000.0',
615        '--hidden-act-deriv-weight', '0.0',
616        '--load-file', 'none'
617    ]
618
619    args4 = [
620        '--task', 'CenterOutAndBackWithFixedGoalReach',
621        # '--file', 'CenterOutAndBackWithFixedGoalReach_pointmass.pt',
622        '--file', 'CenterOutAndBackWithFixedGoalReach_arm.pt',
623        '--epochs', '3000',
624        '--batch-size', '128',
625        # '--effector-type', 'point-mass',
626        '--effector-type', 'arm',
627        '--hidden-sizes', '32', '16',
628        '--lr', '1e-3',
629        '--muscle-weight', '0.0',
630        '--hidden-act-weight', '1000.0',
631        '--hidden-act-deriv-weight', '0.0',
632        '--load-file', 'none',
633    ]
634
635    args5 = [
636        '--task', 'DelayedCenterOutReach',
637        # '--file', 'DelayedCenterOutReach_pointmass.pt',
638        '--file', 'DelayedCenterOutReach_arm.pt',
639        '--epochs', '3000',
640        '--batch-size', '128',
641        # '--effector-type', 'point-mass',
642        '--effector-type', 'arm',
643        '--hidden-sizes', '32', '16',
644        '--lr', '1e-3',
645        '--muscle-weight', '0.0',
646        '--hidden-act-weight', '1000.0',
647        '--hidden-act-deriv-weight', '0.0',
648        '--load-file', 'none',
649    ]
650
651    args6 = [
652        '--task', 'JumpingTargetReach',
653        # '--file', 'JumpingTargetReach_pointmass.pt',
```

```
            '--file', 'JumpingTargetReach_arm.pt',
            '--epochs', '3000',
            '--batch-size', '128',
            # '--effector-type', 'point-mass',
            '--effector-type', 'arm',
            '--hidden-sizes', '32', '16',
            '--lr', '1e-3',
            '--muscle-weight', '0.0',
            '--hidden-act-weight', '1000.0',
            '--hidden-act-deriv-weight', '0.0',
            '--load-file', 'none',
        ]

    args7 = [
            '--task', 'CenterHold',
            '--file', 'CenterHold_pointmass.pt',
            # '--file', 'CenterHold_arm.pt',
            '--epochs', '3000',
            '--batch-size', '128',
            '--effector-type', 'point-mass',
            # '--effector-type', 'arm',
            '--hidden-sizes', '32', '16',
            '--lr', '1e-3',
            '--load-file', 'none'
        ]
    arguments = [args1, args2, args3, args4, args5, args6]
    # args = parser.parse_args(args1)

    print('Running in IPython mode.',file = sys.stdout)
    sys.stdout.flush()
else:
    args = parser.parse_args()
    if rank == 0:
        print('Running in script mode.',file = sys.stdout)

# %% [markdown]
# Extract arguments and train model.

# %%%%
# arguments = [ # this has to come from SLURM script
#     '--effector-type', 'point-mass',
#     '--task', 'RandomTargetReach',
#     '--file', 'RandomTargetReach_pointmass.pt'
# ]
# args = parser.parse_args(arguments)
task = args.task
print("The task inside the main script is now",task, file = sys.stdout)
exp_state = args.exp_state
print('Exp state is set to', exp_state, file = sys.stdout)
epochs = args.epochs
print("The epochs inside the main script now are",epochs,file = sys.
    stdout)
batch_size = args.batch_size
effector_type = args.effector_type
print("The effector type in the main script is ",effector_type,file = sys
    .stdout)
target_radius_frac = args.target_radius_frac
noise_level = args.noise_level
force_field = args.force_field
hidden_sizes = args.hidden_sizes
h0_scale = args.h0_scale
positive_func = args.positive_func
critic_weight = args.critic_weight
muscle_weight = args.muscle_weight
hidden_act_weight = args.hidden_act_weight
```

```
717  print("The_hidden_act_weight_in_the_main_script_is", hidden_act_weight,
         file = sys.stdout)
718  hidden_act_deriv_weight = args.hidden_act_deriv_weight
719  lr = args.lr
720  lr_min = args.lr_min
721  max_grad_norm = args.max_grad_norm
722  rl_for_BG = args.rl_for_BG
723  gamma = args.gamma
724  lmbda = getattr(args, 'lambda')
725  separate_scale = args.separate_scale
726  check_grad = args.check_grad
727  auto_grad_through_rl = args.auto_grad_through_rl
728  rl_lr = args.rl_lr
729  rl_lr_min = args.rl_lr_min
730  print_every = args.print_every
731  save_every = args.save_every
732  file = args.file
733  load_file = args.load_file if args.load_file is not None else file
734  torch_seed = args.seed
735  dtype = torch.float64 if args.dtype == 'float64' else torch.float32
736  device = torch.device(args.device)
737
738  np_seed = torch_seed + 42
739
740
741  for run in range(1):
742      print(f'Starting_{task}_for_{effector_type}_at_{run}_of_1', file =
             sys.stdout)
743      env, actor, critic, optimizer, scheduler, stats = train_model()
744      _save(env, actor, critic, optimizer, scheduler, stats, run)
745
746  print('done', file = sys.stdout)
```

Listing 9: Transfer Learning Main Script

```
1   class ConstrainedGoalReluPointMass(TorchRandomNumberGenerator,
        ReluPointMass24):
2       """
3       Bounds_the_range_of_allowable_goals_and_start_states.
4       """
5       def __init__(self, *args, xy_goal_min=None, xy_goal_max=None, **
            kwargs):
6           super().__init__(*args, **kwargs)
7           self.xy_goal_min = torch.tensor(xy_goal_min, dtype=torch.float32,
                device=self.device)
8           self.xy_goal_max = torch.tensor(xy_goal_max, dtype=torch.float32,
                device=self.device)
9
10      @property
11      def center(self) -> torch.Tensor:
12          if not hasattr(self, '_center'):
13              self._center = 0.5 * (self.xy_goal_max - self.xy_goal_min) +
                    self.xy_goal_min
14          return self._center
15
16      def draw_random_uniform_states(self, batch_size):
17          """Draws_joint_states_according_to_a_random_uniform_distribution,
                _bounded_by_the_position_and_velocity_boundary
18          attributes_defined_at_initialization.
19
20          Args:
21              batch_size:_'Integer',_the_desired_batch_size.
22
23          Returns:
24              A_'tensor'_containing_'batch_size'_joint_states.
```

```python
        """
        sz = (batch_size, self.dof)
        rnd = torch.rand(size=sz, generator=self.th_random, dtype=torch.
            float32, device=self.device)
        pos = (self.xy_goal_max - self.xy_goal_min) * rnd + self.
            xy_goal_min
        vel = torch.zeros_like(pos)
        return torch.cat([pos, vel], dim=1)
```

Listing 10: Effector Class: Point Mass

```python
class ConstrainedGoalRigidTendonArm(TorchRandomNumberGenerator,
    RigidTendonArm26):
    """
    Bounds_the_range_of_allowable_goals_and_start_states._This_requires_
        us_to_solve_an_inverse_kinematics_problem_(i.e.,
    going_from_cartesian_to_joint_space)_which_we_do_numerically.
    """
    def __init__(self, *args, xy_goal_min=None, xy_goal_max=None, c2j_n:
        None | int = None, **kwargs):
        super().__init__(*args, **kwargs)
        self.xy_goal_min = torch.tensor(xy_goal_min, dtype=torch.float32,
            device=self.device)
        self.xy_goal_max = torch.tensor(xy_goal_max, dtype=torch.float32,
            device=self.device)
        if c2j_n is not None:
            self._c2j_n = c2j_n
            self._c2j_xs = torch.linspace(self.xy_goal_min[0].item(),
                self.xy_goal_max[0].item(), c2j_n)
            self._c2j_ys = torch.linspace(self.xy_goal_min[1].item(),
                self.xy_goal_max[1].item(), c2j_n)
            self._c2j_dx = self._c2j_xs[1] - self._c2j_xs[0]
            self._c2j_dy = self._c2j_ys[1] - self._c2j_ys[0]
            xs_grid, ys_grid = torch.meshgrid(self._c2j_xs, self._c2j_ys)
            xy = torch.stack((xs_grid.flatten(), ys_grid.flatten())).T
            self._c2j_thetas = self.cartesian2joint(xy)

    @property
    def center(self) -> torch.Tensor:
        if not hasattr(self, '_center'):
            self._center = self.cartesian2joint(0.5 * (self.xy_goal_max -
                self.xy_goal_min) + self.xy_goal_min[None, :])[0]
        return self._center

    def draw_random_uniform_states(self, batch_size):
        """Draws_joint_states_according_to_a_random_uniform_distribution,
            _bounded_by_the_position_and_velocity_boundary
        attributes_defined_at_initialization.

        Args:
            batch_size:_'Integer',_the_desired_batch_size.

        Returns:
            A_'tensor'_containing_'batch_size'_joint_states.
        """
        max_sample_attempts = 1000

        sz = (batch_size, self.dof)

        # Attempt to sample valid joint positions
        # If the sampling fails, we will retry up to max_sample_attempts
            times
        # If we still cannot sample valid joint positions, we will raise
            an error
        for attempt in range(max_sample_attempts):
```

```
44              rnd = torch.rand(size=sz, generator=self.th_random, dtype=
                    torch.float32, device=self.device)
45              xy_pos = (self.xy_goal_max - self.xy_goal_min) * rnd + self.
                    xy_goal_min
46              try:
47                  pos = self.cartesian2joint(xy_pos)
48                  vel = torch.zeros_like(pos)
49                  return torch.cat([pos, vel], dim=1)
50              except AssertionError:
51                  if attempt == max_sample_attempts - 1:
52                      raise RuntimeError(f'Failed_to_sample_valid_joint_
                            positions_after_{max_sample_attempts}_attempts._
                            Please_check_the_goal_bounds_and_the_
                            cartesian2joint()_implementation.')
53                  continue
54
55
56
57      def cartesian2joint(self, pos: torch.Tensor) -> torch.Tensor:
58          if self.dof != 2:
59              raise NotImplementedError('We_require_2D')
60
61          if pos.requires_grad:
62              raise NotImplementedError('We_currently_cannot_take_gradients
                    _through_cartesian2joint()')
63
64          if hasattr(self, '_c2j_thetas'):
65              i_x = (pos[:, :1] < self._c2j_xs).to(torch.int).argmax(dim
                    =-1)
66              i_y = (pos[:, 1:] < self._c2j_ys).to(torch.int).argmax(dim
                    =-1)
67              w_x = (self._c2j_xs[i_x] - pos[:, 0]) / self._c2j_dx
68              w_y = (self._c2j_ys[i_y] - pos[:, 1]) / self._c2j_dy
69              tmp0 = w_x[:, None] * self._c2j_thetas[(i_x - 1) * self.
                    _c2j_n + i_y - 1] + (1 - w_x)[:, None] * self._c2j_thetas
                    [i_x * self._c2j_n + i_y - 1]
70              tmp1 = w_x[:, None] * self._c2j_thetas[(i_x - 1) * self.
                    _c2j_n + i_y] + (1 - w_x)[:, None] * self._c2j_thetas[i_x
                     * self._c2j_n + i_y]
71              thetas = w_y[:, None] * tmp0 + (1 - w_y)[:, None] * tmp1
72          else:
73              cs: torch.Tensor = (self.skeleton.L2**2 - self.skeleton.L1**2
                     - (pos**2).sum(dim=-1)) / (-2 * self.skeleton.L1)
74
75              theta_param = torch.zeros_like(pos[:, 0]).requires_grad_(True
                    )
76              lbfgs = torch.optim.LBFGS([theta_param], max_iter=100,
                    line_search_fn='strong_wolfe')
77              def closure():
78                  lbfgs.zero_grad()
79                  loss = ((pos[:, 0] * theta_param.cos() + pos[:, 1] *
                        theta_param.sin() - cs)**2).sum()
80                  loss.backward()
81                  return loss
82              lbfgs.step(closure=closure)  # type: ignore
83              theta0 = theta_param.detach()
84
85              theta1 = torch.arctan2(pos[:, 1] - self.skeleton.L1 * theta0.
                    sin(), pos[:, 0] - self.skeleton.L1 * theta0.cos()) -
                    theta0
86              theta1 = (theta1 - self.skeleton.pos_lower_bound[1]) % (2 *
                    torch.pi) + self.skeleton.pos_lower_bound[1]
87
88              thetas = torch.stack((theta0, theta1), dim=-1)
```

```
89          assert (thetas >= self.skeleton.pos_lower_bound).all() and (
                thetas <= self.skeleton.pos_upper_bound).all(), '
                cartesian2joint() optimization failed. Sampled joint position
                 is out of the allowed range'
90          return thetas
```

Listing 11: Effector Class: Arm

