# OpenReview forum: "Effector Complexity Enhances Transfer Learning"
_ICLR.cc/2026/Conference — ICLR 2026 Conference Withdrawn Submission_

### Official Review · Reviewer_bxxD · 2025-10-22

**Soundness:** 1
**Presentation:** 1
**Contribution:** 1
**Rating:** 2
**Confidence:** 3

**Summary:**

The paper aims to study the effect of embodiment complexity and the transfer learning efficiency, hypothesizing that a complex effector provides constraints that lead to better transfer learning. They compare two effectors with different complexity, point mass and arm, and study their transfer learning efficiency on a family of 2D reaching tasks. They use AUC as a major metric to compare the transfer learning efficiency between the tasks.

**Strengths:**

While I am not an expert, the role of embodiment in learning seems to be an overlooked research field that can potentially lead to a lot of interesting discoveries. Particularly, their hypothesis on the complexity of the effector to transfer learning efficiency sounds like an interesting direction.

**Weaknesses:**

1. Limited validation.

The main hypothesis that the authors claim to test is whether the effector complexity provides constraints that yield better transfer learning on new tasks [line 17-18]. I am not convinced that the current experiments are enough to validate the hypothesis.
1) The only compared effectors are two kinds.
I am not suggesting that the authors should look at all types of possible effectors, but I think two is too small a number for the main control variables. Furthermore, it intuitively makes sense that the arm would be a more complex effector, but there is no quantitative measurement of complexity suggested. Since they are comparing only two effectors, it is not convincing yet if these are generalizable results across different effects. Currently, there is only a very qualitative observation across two control variables, that 'arm effector' shows higher transfer efficiency than 'point mass' effector. I think there should be more experiments to draw a conclusion that 'effector complexity' causes better transfer learning.

2) ...'provides constraints that yield better transfer learning' is not validated.
The authors only provide the comparison of performance metrics between two effector types; there is no evidence that this is because the complex effector provides more beneficial constraints.


2. Choice of Metrics.
The authors propose two metrics for analyzing 'learning dynamics': 1) Time to Plateau (TTP) and 2) AUC, which are based on performance measurement of the fraction of time spent in the target zone. The writing does not fully justify the choice of those specific metrics and for example, TTP has multiple hyperparameters of choice (epsilon, delta thresholds) and the authors did not mention the effect or stability of the metrics across the hyperparameter choices.

The authors heavily use AUC metrics, but this metric does not take into account the final accuracy. That is, one model can have a higher AUC but lower final accuracy, and in this case, only looking at AUC won't give a precise comparison of the transfer learning effect.

I think the term 'learning dynamics' is misused because the authors are only measuring efficiency or learning time, rather than other aspects that comprise 'learning dynamics'.




3. Interpretation/communication of results

There are a few points that I do not agree with the author's interpretation and explanations that cause confusion. In section 3.1, there are many parts where the authors use the aggregated term 'performance', but I think it needs a clear distinction between AUC (a proxy measure of learning efficiency) and final performance.

In lines [429-431], the authors claim PM shows negligible/negative transfer, while the ARm model consistently maintains positive transfer. However, this is not true according to Figure 6, as there are tasks where Arm shows negative transfers.

**Questions:**

1. Where is TTP or a measure of task difficulty used?
2. I don't think the experiments were done over multiple seeds?
3. Is the choice of metric (fraction of the time target zone) conventional? What is the justification for this particular metric? Is there any alternative?
4. Could you compare final accuracy, not only AUC, on the transfer task?
5. What drives the observed effect?
The authors only perform validation on purely behavioral and performance-centered metrics. I wonder if authors have looked at the change of representation caused by the effector complexity and how it interacts with the transfer learning task.
I do acknowledge this might not be within the scope of the project, but nevertheless, I believe future work to extend the analysis into a representational level or more quantitative evidence on how affector complexity actually leads to helpful constraint will improve the contribution of the paper significantly. This is a mere suggestion and question out of curiosity, and this is not the main reason for my scoring.

---

> ### Author Response · Authors · 2025-12-03
>
> We thank the reviewer for highlighting the potential importance of embodiment in learning systems. We respond to the raised concerns below.
>
> [1] Number of effectors and generality of conclusions
> The study focuses on a controlled comparison where morphology is the only factor that differs, while simulator, policy architecture, loss function, optimization, and training setup are held constant. This allows us to test whether changing effector complexity alone can influence transfer behavior.
> We agree that extending the analysis to more effectors would strengthen generality, and we now emphasize this direction. However, even with two effectors, the observed transfer pattern appears consistently across six distinct tasks, suggesting that the effect is not tied to a single task or isolated configuration.
>
> [2] Constraints and interpretation of the hypothesis
> The hypothesis is not that complexity always provides better constraints, but that in muscle-based differentiable control systems, a more complex body can provide richer intermediate substrate (e.g., multi-joint synergies, redundancy, and nonlinear actuation mappings) that a pretrained controller may reuse when fine-tuned. We do not claim this as a universal law; the study tests this possibility in a controlled setting.
> We agree that or explicit measures of embodiment constraints would strengthen this connection. These are promising directions for future work but were beyond the scope of the present study.
>
> [3] Quantifying embodiment complexity
> We agree that providing a quantitative complexity measure would make the comparison more explicit. In the current study, the complexity difference is defined operationally through:
> * number of links (1 vs. 2),
> * number of muscles (4 vs. 6),
> * dynamic structure (point dynamics vs. multi-joint articulated dynamics),
> We now clarify this definition more explicitly. A full morphology ladder or complexity continuum is an important next step, but the current setup isolates a meaningful and interpretable contrast.
>
> [4] Choice of metrics: AUC and TTP
> AUC.
> AUC captures learning efficiency (how quickly performance improves), which is central for evaluating transfer. We argue that it is more robust than final accuracy because it evaluates based on speed and peak performance together. It complements final accuracy, particularly for tasks where much of the learning occurs early. Final accuracy can be added as an additional plot, but AUC remains meaningful because transfer can accelerate learning even if final performance is similar.
> TTP:
> TTP is used in Fig. 7 and is defined in Section 2.3 above Equation (1). It serves as a task-difficulty measure rather than a central result metric.
>
> [5] Interpretation of transfer results (lines 429–431)
> As noted above, AUC integrates both speed and accuracy. We acknowledge that speed and accuracy could be analyzed separately as future work.
> Regarding lines 429-431: we thank the reviewer for pointing this out. Arm does show negative transfer on only one task pair (DCOR on COR). The statement in the text refers to the aggregate pattern, not every individual task. We will clarify the wording to avoid overgeneralization.
>
> [6] Multiple seeds
> All experiments are run over 10 random seeds per base task per effector, and each is evaluated on 5 transfer tasks (300 transfer runs per effector and 2 effectors makes it 600 models). We will make this more explicit.
>
> [7] Representation-level analyses
> We appreciate the reviewer’s suggestion. Representation-change analysis is indeed an interesting direction and would complement the behavioral results. This is outside the scope of the present work, but we mention it in the discussion as an avenue for future study.

---

### Official Review · Reviewer_iv76 · 2025-10-30

**Soundness:** 2
**Presentation:** 3
**Contribution:** 1
**Rating:** 0
**Confidence:** 5

**Summary:**

The paper attempt to study the role of embodiment complexity on transfer learning in embodied control tasks. For that a study is designed with 2 different embodiments, a point mass and a 2 link arm and a study is performed on the transfer learning properties of the learned controllers when varying the tasks. The authors propose using the performance area under the curve to compare the different instances. They study their hypotheses using 6 different simulated reaching tasks. The main outcome of the study is twofolds, 1) simpler embodiment leads to favorable learning properties 2) more complex embodiments allow better transfer across tasks.

**Strengths:**

- The paper is well written
- The topic is very interesting, understanding the role of embodiment complexity in task transfer learning can have a huge impact on embodied intelligence
- the results are interesting

**Weaknesses:**

- It is unclear what method was used to learn the controllers
- The experimental setup is very problematic. With only 2 embodiments and one single learning method, the results are not meaningful at all and are hard to map back to specific cause. There are multiple factors that could be leading to this result, such as the choice of the learning algorithm, or the size of the neural network controller. To properly validate the authors' hypotheses a way larger study is required. The least required change is to have a larger number of embodiments, and different learning methods should be studied.
- Given the previously mentioned flaws in the experiment design, setup, and scale, the claims of the paper are wild.

**Questions:**

- can the authors elaborate on the intuition behind their hypothesis that more complex embodiments should have favorable properties in transfer learning? one could easily make the argument that for artificial systems, let's say a neural network controller, the more complex the embodiment the more prone the network is to overfitting and hence transfer is worse...

---

> ### Author Response · Authors · 2025-12-03
>
> We thank the reviewer for the positive comments on the writing, topic, and their interest in the results. We address the concerns below.
>
> [1] Clarity about methods used: We apologize for the lack of clarity. As described in Sec. 2.2 and shown in Appendix A.2, controllers are trained with supervised differentiable control using a GRU policy, with gradients propagated through the differentiable muscle-based simulation (Motornet).
>
> [2] Experiment setup: We agree that a larger-scale exploration of embodiment space would be valuable. Our aim in this paper is more modest: to test whether varying effector complexity alone, under fully controlled conditions (same simulator, same GRU policy, same loss, same optimization), can influence transfer behavior. Even with only one simple and one more complex effector, the design is a controlled comparison, not a random sample. Morphology is the only factor that differs, which allows us to isolate the effect of effector complexity cleanly. In addition, the effect is consistent across six tasks with different kinematics, delays, and target structures. If the result were driven by an idiosyncrasy of a single effector, we would not expect the same transfer pattern to appear across all six task families.
> While we agree that extending this to more effectors and additional learning methods is an important next step, this does not, in our view, invalidate the current findings.
>
> [3]: Strength of claims: We appreciate the opportunity to clarify. Our claims are intentionally limited:
> we do not claim a universal advantage of complex embodiments;
> we do not claim general laws of embodiment;
> we only claim that, in this controlled setup, a more complex effector shows larger transfer gains across six tasks;
> and that a simpler effector learns faster from scratch.
>
> [4] Intuition: The intuition we had is not that more complex embodiments should universally produce better transfer, but that they may offer more ways to solve a new task when fine-tuned, because their mapping from muscle activations to movement is higher-dimensional and more flexible. In other words, a complex effector can express a richer set of movement primitives and coordination strategies, which might make it easier for a pretrained controller to adapt or recombine existing patterns when a new task is introduced.
>
> At the same time, we agree with the reviewer that increased complexity can also create opportunities for overfitting or greater sensitivity to the training procedure. Our hypothesis was therefore not that complexity always helps transfer, but that it could do so under certain conditions. The present study tests this possibility in a controlled setting.

---

### Official Review · Reviewer_MJyL · 2025-10-31

**Soundness:** 2
**Presentation:** 2
**Contribution:** 3
**Rating:** 2
**Confidence:** 4

**Summary:**

This paper investigates how different embodiments of end effectors affect transfer learning abilities. The paper compares a point end effector and an arm across 6 different tasks  The paper hypothesizes that more complex embodiments improve transfer learning abilities and show results indicating that this might be the case

**Strengths:**

The paper poses an interesting hypothesis that could have important implications in robotics and transfer learning. The authors test their hypothesis across 6 different tasks and show consistent results across these 6 tasks.

**Weaknesses:**

It would be helpful to average figure 2 across all tasks. This would make the advantage of the transfer model more apparent.
The authors never stated that this was done in simulation. It would be useful to state this up front.
Fig 3 could be more clearly labeled. what is the orange circle? where does the robot start?
I would have thought that the arm would take longer to train considering the multiple degrees of freedom - seems like there is little difference. Could the authors comment on this?
How many seeds was this evaluated over?
Fig 4 should show standard error . Also its unclear what the red/orange bar means on fig 4 g

**Questions:**

The biggest issue with the paper is that it is not clear if the results were produced for multiple multiple trials (e.g., multiple random seeds) and as such i am not convince of the significance of the results. The authors state that these gains are all statistically significant but do not discuss statistical tests or include p values anywhere in the main paper.


To better support the hypothesis put forward in this paper, i would like to see results across several simple and several complex embodiments. Only one of each type of embodiment isn't enough to suggest that this finding is true for the reasons hypothesized. It could be do to other factors unrelated to complexity. If this could be done, I would be willing to raise my score.

---

> ### Author Response · Authors · 2025-12-03
>
> We thank the reviewer for the constructive feedback and for noting the consistency of the results across six tasks. We address the points of confusion below.
>
> [1] Averaging Figure 2 across tasks:
> We agree that averaging Fig. 2 across tasks would help illustrate overall trends. In our current analysis, we plot each task separately to preserve task-specific dynamics, but we acknowledge that including an aggregated version would make transfer benefits more visually apparent.
>
> [2] Clarifying that experiments are in simulation.
> All experiments are performed in a differentiable biomechanical simulation environment (Motornet). We agree that stating this explicitly earlier would improve clarity, although Motornet is itself a simulator and the methods section consistently refers to simulated muscle activations and differentiable dynamics.
>
> [3] Clarifying Fig. 3: target and starting position.
> The orange circle represents the target, and the start of the trajectory indicates the initial position. This is already mentioned in the caption.
>
> [4] Training time differences between the arm and point mass.
> The point mass does generally learn faster from scratch than the arm, and this can be seen in Fig. 4 across multiple tasks. This is true for most tasks.
>
> [5] Number of seeds.
> All base-task results are averaged over 10 random seeds per task and effector, producing 60 trained models. Each of these is then fine-tuned on 5 transfer tasks, resulting in 300 transfer runs per effector. With 2 effectors that’s 600 models. We agree that describing this more explicitly in the main text would improve clarity.
>
> [6] Fig. 4g: standard error and color bars.
> Standard error bars are included in Fig. 4g. The red/orange bars represent the AUC (Unit^2) values, as indicated in the accompanying scale in the figure.
>
> [7] Statistical significance and p-values
> Statistical significance was assessed through paired comparisons across the 10 seeds. The number of significant pairs are addressed in Fig 8 but we agree that explicitly stating the statistical test and providing p-values in the main text would improve transparency.
>
> [8] Request for multiple embodiments
> We agree that exploring a broader set of simple and complex embodiments would strengthen the generality of the hypothesis. Our objective in this study was more modest: to test, in a controlled setting, whether varying effector complexity alone can influence transfer when all other factors (policy architecture, optimizer, loss, simulator) are held constant.

---

### Official Review · Reviewer_Rvmo · 2025-11-01

**Soundness:** 2
**Presentation:** 3
**Contribution:** 3
**Rating:** 4
**Confidence:** 3

**Summary:**

The paper asks whether the body an agent controls, its effector, shape how well it can transfer knowledge across tasks? Using differentiable biomechanical simulators (Motornet) and GRU-based policies trained with supervised, differentiable control (not RL), the authors compare a simple 2‑D point mass with four “muscles” to a more complex two‑link, six‑muscle arm across six reaching tasks. From-scratch learning favors the simple effector, but after pretraining on a different task, the complex arm gains more from transfer and sometimes surpasses the point mass in absolute performance.

**Strengths:**

- Clear experimental question and setup. The paper isolates effector complexity as a variable and holds policy architecture largely constant. The training protocol, base model, fine‑tune on target vs. train from scratch is well illustrated (Fig. 2, p. 3).

- The finding that the complex arm benefits more from transfer and can outperform the simpler effector after transfer on several tasks, is interesting.

**Weaknesses:**

- The study’s comparison is restricted to a single embodiment pair; a simple 2‑D point mass actuated by four “muscles” versus a two‑link arm with six muscles without alternative “complex” bodies. Accordingly, the claim that effector complexity enhances transfer would be more robust if the same pattern were replicated across a morphology ladder (e.g., increasing links, actuator counts, or compliance)

- Task budgets and settings vary widely (e.g., 35k epochs for COBFGR and DCOR with different lrs/batch sizes vs. 5k elsewhere). This complicates cross‑task difficulty comparisons (Fig. 7, left) and AUC magnitudes. A control with matched budgets would strengthen claims.

- Loss‑term ablations are missing. Training uses substantial hidden‑activity and muscle‑activation regularization. Without ablations, it remains unclear whether these regularizers, not morphology, explain the arm’s transfer advantage.

**Questions:**

See weakness section

---

> ### Author Response · Authors · 2025-12-03
>
> We thank the reviewer for the clear summary, positive assessment of the experimental question and setup, and for highlighting the interest of the finding that a more complex effector benefits more from transfer. Below we address each concern in turn.
>
> [1] Limited Morphologies: We agree that evaluating a broader set of morphologies would further strengthen the study, and we now explicitly highlight this as a direction for future work. Our goal in this work is narrower: to test whether increasing effector complexity can influence transfer performance in a controlled and well-specified setting. For this purpose, we selected one simple and one more complex effector, both implemented in the same differentiable biomechanics framework and trained under identical architectures and losses, so that morphology is the only varying factor.
> Due to practical constraints (training 2 effectors across 6 base tasks × 5 transfer tasks with 10 seeds each using differentiable muscle-based simulation is computationally intensive), we focused on this pair rather than running a full morphology sweep. Even within this limited set, the same transfer pattern appears consistently across all six tasks, indicating that the effect is robust rather than task-specific.
>
> [2] Difference in training budgets: Tasks in our benchmark differ substantially in structure and difficulty. Some (e.g., COR) converge reliably within ~5k epochs, whereas others with delayed feedback or more complex trajectories (e.g., COBFGR, DCOR) require significantly longer training to reach stable behavior. For this reason, we used task-specific budgets, but critically, within each task, both effectors and both training regimes (Transfer and No-Transfer) use the exact same epochs, learning rate, and batch size. So we compare a transfer learnt model of COR, with a non-transfer learnt model of COR, both of which take the same 5k epochs, learning rate, batch size and loss function.
>
> [3] Regularizers potentially explaining results:  Both effectors are trained with the same loss function and the same regularization weights for the same tasks. Regularization is task specific, not effector specific. Which means that the point-mass and the arm use identical coefficients for the hidden-activity penalty, the muscle-activation penalty, and all other loss components when doing identical tasks. Transfer and No-Transfer conditions also use the same loss. Because the regularization terms are applied equally to both bodies, they may not in our opinion, by themselves, explain why the arm consistently shows larger transfer gains across all six tasks.

---

### Note · Authors · 2025-12-03

I have read and agree with the venue's withdrawal policy on behalf of myself and my co-authors.